# HBZ upregulates myoferlin expression to facilitate HTLV-1 infection

**Nicholas Polakowski[1], Md Abu Kawsar Sarker[1], Kimson Hoang[1], Georgina Boateng[1], Amanda W. Rushing[2], Wesley Kendle[1], Claudine Pique[3,4,5], Patrick L. Green[6], Amanda R. Panfil[6], Isabelle Lemasson[1]***

1 Brody School of Medicine, Department of Microbiology and Immunology, East Carolina University, Greenville, North Carolina, United States of America, 2 Catawba College, Department of Biology, Salisbury, North Carolina, United States of America, 3 INSERM, U1016, Institut Cochin, Paris, France, 4 CNRS, UMR8104, Paris, France, 5 Université Paris Descartes, Sorbonne Paris Cité, Paris, France, 6 Center for Retrovirus Research and Department of Veterinary Biosciences, College of Veterinary Medicine, The Ohio State University, Columbus, Ohio, United States of America

* lemassoni@ecu.edu

**Data Availability Statement:** All relevant data are within the manuscript and its Supporting Information file.

**Funding:** Funding was provided by the National Institute of Health through grants

## Abstract

The complex retrovirus, human T-cell leukemia virus type 1 (HTLV-1), primarily infects CD4$^+$ T-cells *in vivo*. Infectious spread within this cell population requires direct contact between virally-infected and target cells. The HTLV-1 accessory protein, HBZ, was recently shown to enhance HTLV-1 infection by activating intracellular adhesion molecule 1 (ICAM-1) expression, which promotes binding of infected cells to target cells and facilitates formation of a virological synapse. In this study we show that HBZ additionally enhances HTLV-1 infection by activating expression of myoferlin (MyoF), which functions in membrane fusion and repair and vesicle transport. Results from ChIP assays and quantitative reverse transcriptase PCR indicate that HBZ forms a complex with c-Jun or JunB at two enhancer sites within the *MYOF* gene and activates transcription through recruitment of the coactivator p300/CBP. In HTLV-1-infected T-cells, specific inhibition of MyoF using the drug, WJ460, or shRNA-mediated knockdown of MyoF reduced infection efficiency. This effect was associated with a decrease in cell adhesion and an intracellular reduction in the abundance of HTLV-1 envelope (Env) surface unit (SU) and transmembrane domain (TM). Lysosomal protease inhibitors partially restored SU levels in WJ460-treated cells, and SU localization to LAMP-2 sites was increased by MyoF knockdown, suggesting that MyoF restricts SU trafficking to lysosomes for degradation. Consistent with these effects, less SU was associated with cell-free virus particles. Together, these data suggest that MyoF contributes to HTLV-1 infection through modulation of Env trafficking and cell adhesion.

## Author summary

Human T-cell leukemia virus type 1 (HTLV-1) infection can cause a progressive neuroinflammatory disease and a fatal form of leukemia and is also linked to other inflammatory-based pathologies. These clinical manifestations stem, in part, from viral spread within the

R21AI166077and R15AI133412 to IL and P01CA100730 to PLG. The funders had no role in the study design, data collection and analysis, the decision to publish, or preparation of the manuscript.

**Competing interests:** The authors have declared that no competing interests exist.

CD4$^+$ T-cell population of the host. Once infected with HTLV-1, a T-cell adopts a novel gene expression program that renders it competent for infecting other T-cells. This reprogramming event relies on the functions of specific HTLV-1 proteins. Here, we show that induction of myoferlin (MyoF) expression by the viral protein HBZ is important for HTLV-1 infection. Our data support a model in which HBZ activates transcription of the *MYOF* gene by associating with c-Jun or JunB at two sites within the gene and recruiting the cellular coactivator p300/CBP to these sites. We found that, in connection with its known role in vesicle trafficking, MyoF reduces transfer of the HTLV-1 envelope (Env) protein to lysosomes for degradation, leading to more Env incorporation into virions. Additionally, we found that MyoF enhances the adhesion properties of the HTLV-1-infected T-cells, which is important, as cell-to-cell contact is essential for viral infection. Our study provides new insights into the events required for HTLV-1 infection.

## Introduction

It is estimated that approximately 5–10 million people worldwide are infected with the complex retrovirus, human T-cell leukemia virus type 1 (HTLV-1) [1]. A small percentage of this population will ultimately present with clinical manifestations associated with the viral infection. Specifically, HTLV-1 is the etiologic agent of a fatal malignancy known as adult T-cell leukemia (ATL) and, separately, a progressive neurodegenerative disease known as HTLV-1-associated myelopathy/tropical spastic paraparesis (HAM/TSP) [2–5]. In addition, HTLV-1 infection is associated with uveitis, infective dermatitis and linked to other inflammatory-mediated disorders [6–8].

*In vivo*, HTLV-1 primarily infects CD4$^+$ T-cells [9], and infectious spread within this cell population requires contact between virally-infected and target T-cells [10–12]. Initiation and stabilization of these contacts appears to occur primarily through an interaction of intercellular adhesion molecule 1 (ICAM-1) on infected cells with its receptor, leukocyte function associated antigen-1 (LFA-1), on target cells [13]. Virus particles are then transferred to targets cells either through a virological synapse (VS) [14], or from adhesive extracellular biofilm-like viral assemblies [15], or through cellular conduits [16]. In addition to mediating cell adhesion, engagement of ICAM-1 on the surface of infected cells is essential for formation of the VS [17,18].

HTLV-1 envelope protein (Env) is present at the surface of virus particles and is essential for viral infection. The mature form of Env is composed of an external surface unit, SU (also designated gp46), disulfide-bonded to the transmembrane subunit, TM (gp21) [19]. Virus particles dock to target cells initially through interactions of SU with heparan sulfate proteoglycans, and then establish stable cellular contact when SU binds neuropilin-1 [20,21]. Subsequent binding of SU to glucose transporter 1 is believed to induce a conformational change in TM that activates fusion of the viral envelope with the plasma membrane [22,23], leading to viral entry. HTLV-1 Env is highly fusogenic, and thus, high levels of Env at the plasma membrane cause contacted cells to form syncytia [24–26]. This effect is moderated by rapid clathrin-mediated endocytosis of Env, which occurs primarily through interactions between the YXXΦ motif in the cytoplasmic domain of TM (CD-TM) and adaptor protein 2 (AP-2) [27,28]. Interestingly, the YXXΦ motif may also promote lysosomal degradation of Env, a process that appears to be offset by a separate PDZ-binding motif in CD-TM [29]. This latter domain was initially identified based on an interaction with the PDZ domain-containing scaffolding protein, hDLG1, which facilitates accumulation of Env and Gag to discrete sites in proximity to the plasma membrane [30].

The viral protein HBZ was recently shown to enhance HTLV-1 infection by activating ICAM-1 expression [31]. HBZ regulates transcription of many other cellular genes as well as transcription of the HTLV-1 provirus [32]. At its C-terminus, HBZ harbors a leucine zipper (ZIP) domain that dimerizes with several cellular basic leucine zipper (bZIP) transcription factors. In some contexts, these interactions allow HBZ to associate with the DNA, either directly or indirectly. Direct DNA-binding is supported by biochemical evidence indicating that heterodimers formed between HBZ and small MAFs (musculoaponeurotic fibrosarcoma proteins) bind Maf recognition elements (MAREs) [33,34]. Indirect association with the DNA has been illustrated by findings that JunD is able to bridge HBZ to the DNA by simultaneously forming a heterodimer with HBZ and interacting with DNA-bound Sp1 [35,36]. The N-terminal region of HBZ contains an activation domain that forms high-affinity interactions with the paralogous cellular coactivators, p300 and CBP (singularly referred to as p300/CBP) [37,38]. These proteins serve as scaffolds for the recruitment of additional transcriptional regulators and have lysine acetyltransferase (KAT) activity that modify histones as well as transcriptional regulators [39]. Therefore, when associated with the DNA, HBZ may activate transcription through recruitment of p300/CBP.

In this study, we show that HBZ provides an additional contribution to the HTLV-1 infection process by directly activating expression of the *MYOF* gene. This gene encodes myoferlin (MyoF), which is involved in membrane fusion and repair and in vesicle transport [40]. HBZ, in conjunction with Jun family members, was enriched at two sites within the *MYOF* gene positioned approximately 19 and 103 kb downstream of the transcription start site (TSS). We provide evidence that HBZ recruits p300/CBP to these sites and that the coactivator KAT activity is important for *MYOF* transcription. A recently developed specific inhibitor of MyoF known as WJ460 [41] reduced HTLV-1 infection, which was correlated with a reduction in the intracellular level of mature Env. These effects were corroborated by shRNA-mediated knockdown of MyoF. In conjunction with the lower abundance of Env in MyoF knockdown cells, less Env was incorporated into virus particles produced by these cells. One role of MyoF is to regulate endosomal trafficking [42–45]. We provide evidence that this function controls the abundance of mature Env, specifically by restricting trafficking of Env to lysosomes for degradation.

## Results

### Myoferlin is expressed in HTLV-1 infected cells

To gain a clearer insight into how HBZ enhances HTLV-1 infection, we cross-referenced genes that are activated by HBZ and express proteins with functions that might relate to the infection process. We previously performed a gene expression microarray analysis, comparing clonal HeLa cell lines expressing HBZ or carrying the empty expression vector [46,47]. *MYOF*, which encodes the protein, myoferlin (MyoF), was among the genes with higher expression in the presence of HBZ. Although MyoF plays an essential role in myoblast fusion [42], it has also been reported to function in endosomal recycling [42–45,48], a process often expropriated by retroviruses and other viruses [49–51]. Using quantitative reverse transcriptase PCR (qRT-PCR) and western blot analysis, we verified that *MYOF* expression is upregulated in the presence of HBZ (Fig 1A and 1B). In contrast to HBZ, the HTLV-1 encoded protein, Tax, which also acts as a transcriptional regulator, did not affect MyoF expression when expressed in HeLa cells (S1 Fig).

To determine whether HTLV-1-infected T-cells express *MYOF*, we first performed a qRT-PCR analysis across a panel of T-cell lines. We found that levels of *MYOF* mRNA were higher in HTLV-1-infected T-cell lines compared to uninfected T-cell lines; the latter set

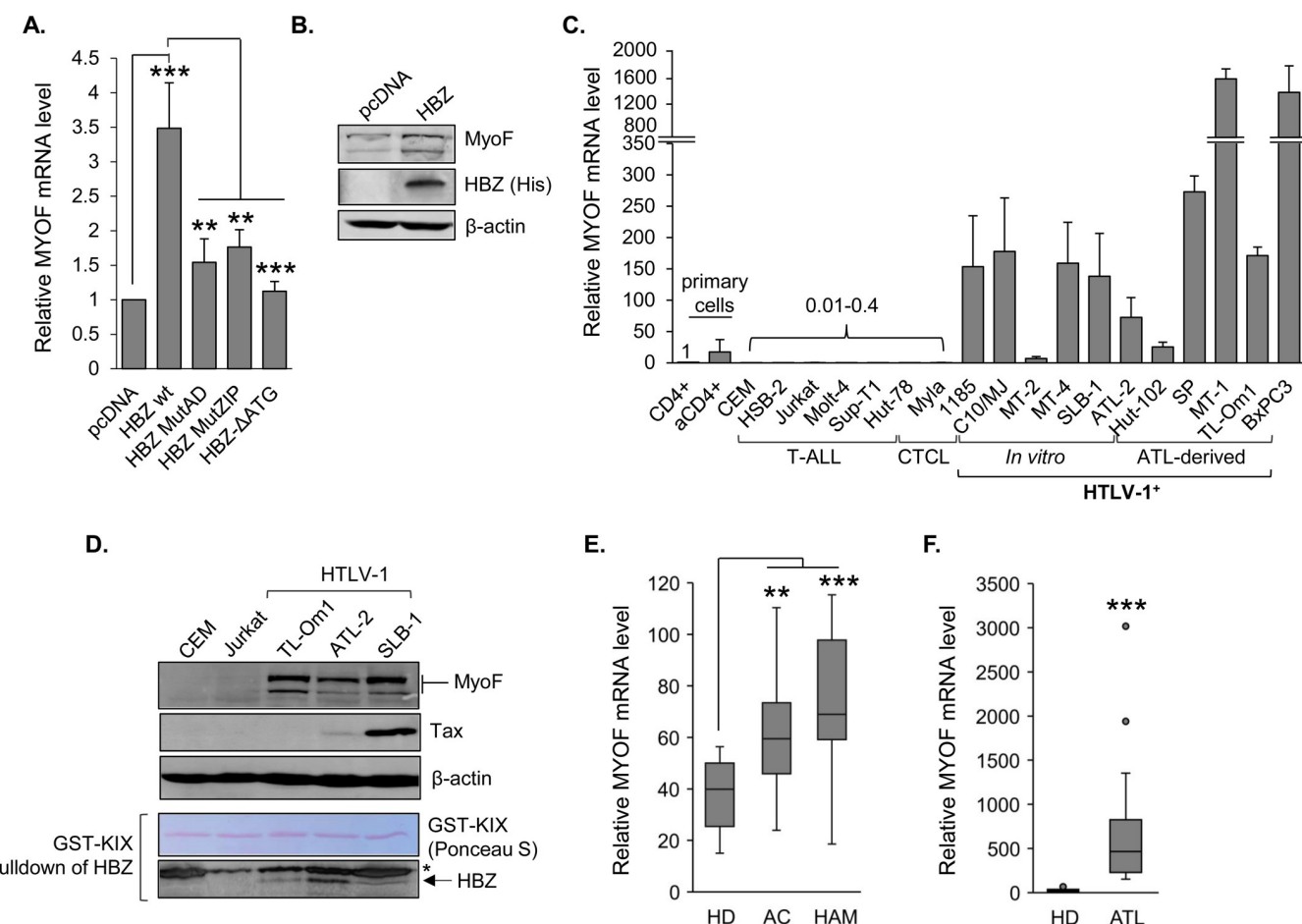

**Fig 1. MyoF is expressed in HBZ-expressing cells and in HTLV-1-infected T-cells. (A)** Relative *MYOF* mRNA levels in the indicated HeLa cell lines. The graph shows qRT-PCR results averaged from four independent experiments with values normalized to that of the empty-vector clone, pcDNA (set to 1), for each replicate. Error bars show standard deviations; ** $p<0.01$, *** $p<0.001$. **(B)** MyoF expression in empty vector (pcDNA) and HBZ-HeLa clones. Whole cell extracts (50 μg) were analyzed by Western blot using antibodies against MyoF, HBZ (6xHis) and β-actin. **(C)** Relative *MYOF* mRNA levels in resting and activated primary CD4+ T-cells and in T-cell lines. The graph shows qRT-PCR results averaged from two independent experiments in which values were normalized to those for the resting CD4+ T-cells (set to 1), error bars show standard deviations. Numbers above the bars indicate expression levels relative to the resting CD4+ T-cells. HTLV-1-negative lines are T-cell acute lymphoblastic leukemia lines (CEM, HSB-2, Jurkat, Molt-4 and SupT1) and cutaneous T-cell leukemia lines (HUT-78 and Myla); *in vitro* transformed HTLV-1-positive cells are 1185, C10/MJ, MT-2, MT-4 and SLB-1; ATL-derived cell lines are ATL-2, HUT-102, SP, MT-1 and TL-Om1. aCD4+ is activated CD4+ T-cells. BXPC3 is a pancreatic cancer cell line. **(D)** MyoF expression in T-cell lines. Whole cell extracts (45 μg) were analyzed by Western blot using antibodies against MyoF, Tax, β-actin and HBZ. Endogenous HBZ was enriched by affinity precipitation using GST-KIX (Ponceau S-stained). **(E)** Tukey boxplots (showing *MYOF* transcript levels partitioned by healthy donor (HD), asymptomatic carrier (AC), HTLV-1 associated myelopathy (HAM) and **(F)** adult T-cell leukemia (ATL) from published microarray data: GEO accession numbers GSE29312 [103] and GSE14317 [104] for (E) and (F), respectively; boxplot ends of each whisker are set to 1.5 times the interquartile range above the third quartile and below the first quartile; ** $p<0.01$, *** $p<0.001$.

containing barely detectable levels of *MYOF* mRNA (Fig 1C). In this analysis, BxPC3 pancreatic cancer cells served as a positive control, as these cells display high *MYOF* expression [52]. We also analyzed *hbz* mRNA copy numbers in one set of RNA from the HTLV-1-infected T-cells lines (S2A Fig) but did not observe a significant correlation between *hbz* mRNA copy number and relative *MYOF* mRNA (S2B Fig). Surprisingly, a comparison of *tax* mRNA copy number to relative *MYOF* mRNA showed a negative correlation (S2C and S2D Fig). It is possible that these long-established cell lines harbor heterogenous sets of mutations that influence *MYOF* transcription independently of the viral transcription factors. Using a subset of HTLV-1-infected cell lines, we verified expression of the MyoF protein (Fig 1D). Consistent with our

qRT-PCR results, analyses of microarray data sets from the Gene Expression Omnibus (GEO) repository revealed significantly higher *MYOF* expression in CD4+ T-cells from asymptomatic HTLV-1 carriers (AC) and HAM/TSP patients (HAM) compared to those from uninfected healthy donors (HD), and significantly higher *MYOF* expression in CD4+ T-cells from ATL patients compared to HDs (Fig 1E and 1F).

## HBZ activates Myoferlin expression

In addition to examining levels of *MYOF* mRNA in empty vector and HBZ-expressing HeLa cells, we examined transcript levels in HeLa cell lines expressing HBZ with mutations in its start codon (HBZ-ΔATG) and its two primary transcriptional regulatory domains: the N-terminal activation domain (HBZ-mutAD) and the C-terminal leucine zipper domain (HBZ-MutZIP). Mutating the HBZ start codon abolished *MYOF* induction, demonstrating that the HBZ protein rather than the transcript activates *MYOF* expression (Fig 1A). Moreover, the effect of the two other mutations indicates that both transcriptional regulatory domains of HBZ are required for activation. Supporting a role for only HBZ in activating *MYOF* expression in HTLV-1-infected T-cells, we showed that expression of the mRNA and the MyoF protein was present in TL-Om1 cells (Fig 1C and 1D). These cells express HBZ but do not express other HTLV-1 proteins due to the methylation pattern of the provirus [53–55]. We also analyzed GEO microarray data sets from ATL cell lines (KK1 and ST1) that were modified by CRISPR/Cas9 to knock out *hbz* [56]. The two guide RNAs successfully targeting *hbz* led to a reduction in *MYOF* mRNA in both cell lines (Fig 2A).

To confirm the involvement of HBZ in *MYOF* expression, we analyzed *MYOF* mRNA levels in recently established HTLV-1-immortalized clones from human peripheral blood lymphocytes (PBLs). Clones were established using either 729.HTLV-1 cells that carry the wild-type (wt) HTLV-1 molecular clone, ACH, or using 729.HTLV-1ΔHBZ cells in which a point mutation in ACH produces a stop codon terminating translation of the major splice variant of HBZ at amino acid 8 [57]. *MYOF* mRNA was detected in all clones albeit at varying levels, which may relate to the use of PBLs from different donors for establishing these clones (Fig 2B, dark grey bars). Compared to the four clones harboring the wild-type virus, the two clones with the ΔHBZ virus displayed lower levels of *MYOF* mRNA (Fig 2B, hatched bars). This trend was more uniform among clones established from the same donor (Fig 2B, black bars). Levels of *tax* and *hbz* mRNA relative to those in SLB-1 cells were also determined for each clone (Fig 2C). Interestingly, levels of *hbz* mRNA were lower in clones harboring the ΔHBZ virus, possibly due to the lack of a positive feedback loop by the HBZ protein on its own promoter [35]. In comparing relative *hbz* and *MYOF* mRNA levels, we observed a significant correlation; however, there was no correlation between *tax* and *MYOF* mRNA levels (S3 Fig), confirming that Tax is not involved in *MYOF* induction. These results support the role of HBZ in regulating *MYOF* transcription and also support the premise that *MYOF* expression arises from the viral infection rather than from the evolution of long-established HTLV-1-infected T-cell lines.

## HBZ binds to the *MYOF* gene in conjunction with cellular AP-1 transcription factors

To determine how HBZ activates *MYOF* expression, we first analyzed GEO ChIP-seq data sets from the Nakagawa *et al.* study in which ectopically-expressed, biotin-tagged HBZ was used to precipitate chromatin [56]. Two peaks of HBZ enrichment were present within the *MYOF* gene: one in the 5' region of the gene close to the promoter [~19 kb from transcription start site (TSS); proximal peak] and another downstream in the gene (~103 kb from TSS; distal peak; Fig 3A). Given this observation, we performed ChIP assays comparing the empty vector

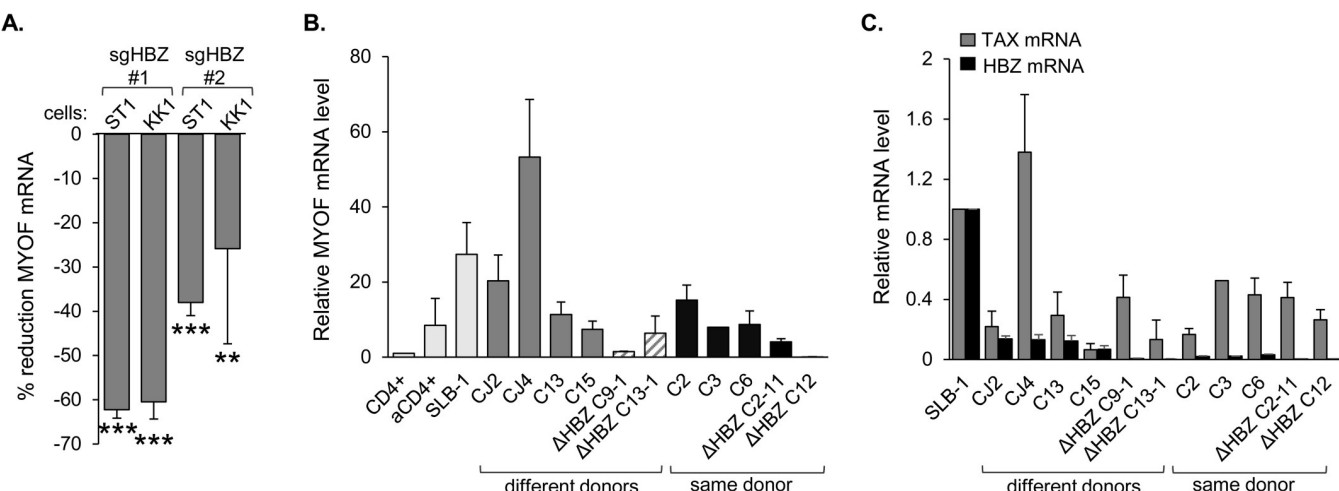

**Fig 2. Expression of HBZ correlates with MyoF expression. (A)** Deletion of HBZ in ST1 and KK1 ATL cells reduces *MYOF* expression. The graph was generated from published microarray data (GEO accession number GSE94409 [56]) and shows the percent reduction in *MYOF* transcript levels seven days after inducing CRISPR/Cas9-mediated knockout of HBZ in the ATL cell lines, ST1 and KK1, using two different guide RNAs (sgHBZ #1 and #2) compared to the guide RNA control. Data were obtained using GEO2R with calculations based on averaged values from the three array features probing for different regions of the MyoF transcript; ** $p<0.01$, *** $p<0.001$. **(B)** Relative *MYOF* mRNA levels in HTLV-1-immortalized human T-cell clones recently established from peripheral blood lymphocytes (PBL). The graph shows qRT-PCR results as follows: C3, value from one RNA extraction; ΔHBZ clones, values averaged from two extractions; all others, values averaged from three extractions. Values were normalized to those for the resting CD4+ T-cells (set to 1). Dark gray bars show clones derived from different PBL donors immortalized with HTLV-1-wt (CJ2, CJ4, C13, C15). Hatched bars show clones derived from different PBL donors immortalized with HTLV-1-ΔHBZ (C9-1 and C13-1). Black bars show clones immortalized with either HTLV-1-wt or HTLV-1-ΔHBZ derived from the same PBL donor (C2, C3, C6, C2-11 and C12). Light gray bars show a subset of cells also analyzed in Fig 1C for comparison. **(C)** Relative *tax* and *hbz* mRNA levels in HTLV-1-immortalized human T-cell clones recently established from PBL. The graph shows qRT-PCR results averaged from the same set of RNA specimens analyzed in (B) in which values were normalized to those for the SLB-1 T-cells (set to 1).

and HBZ-expressing HeLa cell lines and using the C-terminal 6xHis epitope tag on HBZ for immunoprecipitation [46]. Significant enrichment of HBZ was observed at the proximal peak region, but not the distal peak region (Fig 3B). Because the chromatin environment potentially varies between HeLa cells and HTLV-1-infected T-cells, we performed additional ChIP assays using transduced SLB-1 cells expressing HBZ with a C-terminal 6xHis tag for immunoprecipitation. Compared to the negative control off-target site, a two-fold increase in HBZ enrichment was observed at both the proximal and distal peaks (Fig 3C). This result is consistent with the *in-silico* analysis.

The DNA sequences corresponding to the proximal and distal peaks were found to contain full or partial AP-1-binding sites (sequences in S4 Fig). HBZ is known to interact with AP-1 members, such as c-Jun, JunB and JunD [33,58, 59]. However, when associated with HBZ, these factors are generally inhibited from binding AP-1 sites and activating transcription [32]. In contrast, complexes formed by HBZ and small Mafs bind MAREs, which contain a core AP-1-binding site sequence and activate transcription [34]. To determine whether any of these cellular bZIP factors are associated with either of the two HBZ-binding peaks, we performed ChIP assays in ATL-2 cells. Interestingly, we observed significant enrichment of JunB and c-Jun at both the distal and proximal peaks (Fig 3D and 3E, respectively). Although levels of JunD enrichment at these sites appeared higher than that of the off-target site, values were not significant (Fig 3F). This outcome may have been influenced by the antibody efficiency; the JunD antibody produced a lower positive control signal (WEE1-AP1) than the other antibodies. No detectable enrichment of MafG was observed at either the distal or proximal peaks (Fig 3G). These results suggest that HBZ is recruited together with, or cooperates with at least c-Jun or JunB on the *MYOF* gene.

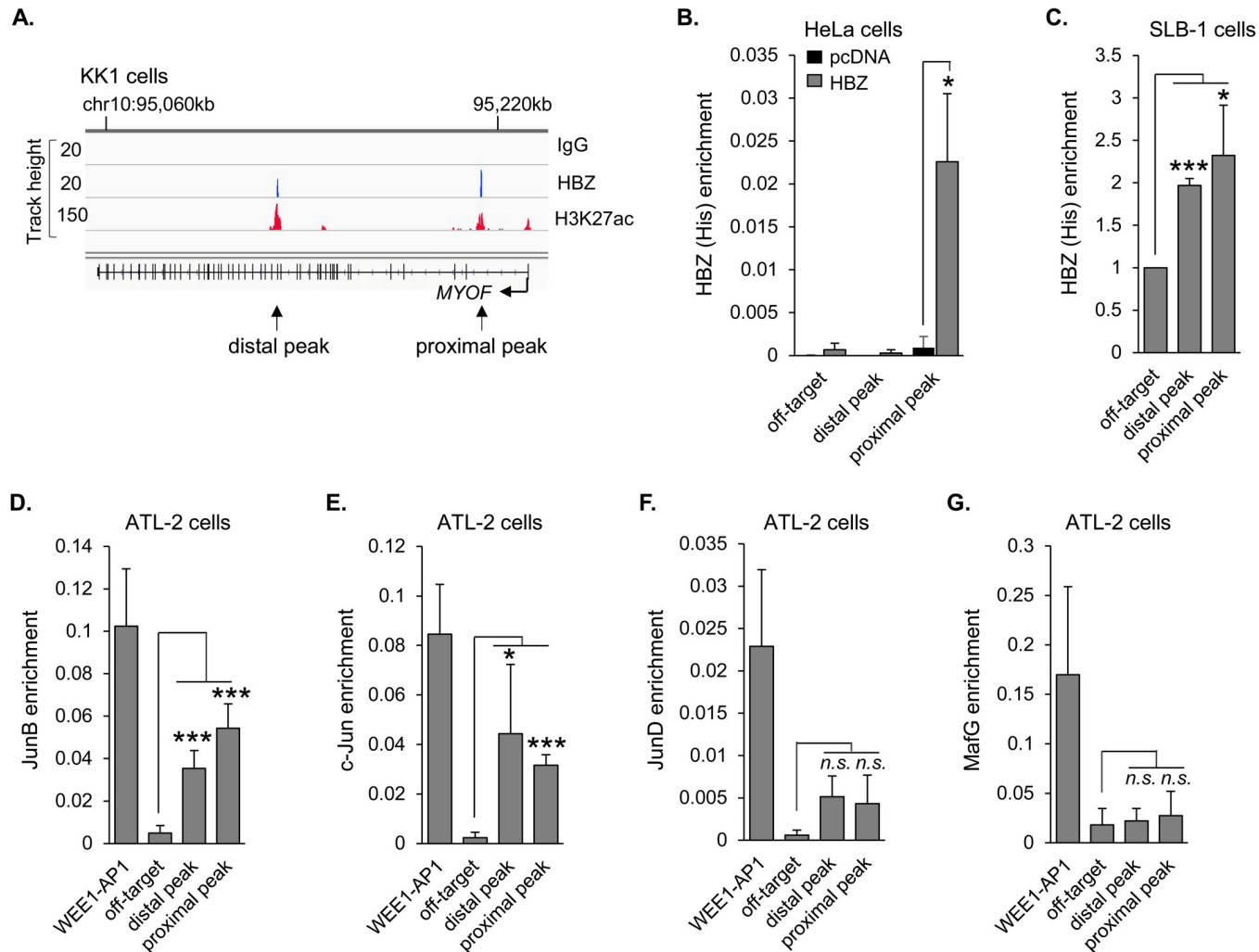

**Fig 3. HBZ binds the *MYOF* gene in conjunction with c-Jun and JunB. (A)** HBZ binds to two sites within the *MYOF* gene in the ATL cell line, KK1. Peaks of enrichment for HBZ, the epigenetic mark, H3K27ac, and control (IgG) at the *MYOF* locus in KK1 cells are shown in the IGV Browser. Genomic coordinates are based on the NCBI36/hg18 assembly. Data were obtained from published ChIP-Seq data sets (GEO accession number GSE94732 [56]). **(B)** HBZ binds to the proximal peak in HeLa cells. The graph shows average values from three independent ChIP assays using empty vector (pcDNA) and HBZ (6x His C-terminal tag)-expressing HeLa cells. Shown are levels of HBZ-enrichment at the off-target site (1,796 bp from the end of the gene) and at the distal and proximal peaks with respect to the TSS; * $p<0.05$. **(C)** HBZ binds the distal and proximal peaks in SLB-1 cells. The graph shows fold enrichment relative to that for the off-target site (set to 1) averaged from three independent ChIP assays using SLB-1 cells transduced to express HBZ with a C-terminal 6x His tag; * $p<0.05$, *** $p<0.001$. **(D)** and **(E)** JunB and c-Jun are enriched at the distal and proximal peaks in ATL-2 cells. **(F)** and **(G)** JunD and MafG are not enriched at the HBZ-binding peaks. Graphs show average values from five (D and E) to three (F and G) independent ChIP assays using ATL-2 cells; * $p<0.05$, *** $p<0.001$, *n.s*: non-significant. Factor-enrichment was analyzed at the off-target site, the distal and proximal *MYOF* peaks, and at the AP1 site in the *WEE1* promoter (WEE1-AP1).

## HBZ activates *MYOF* transcription through recruitment of p300/CBP

Analysis of GEO ChIP-seq data sets from the Nakagawa study also revealed peaks of enrichment for histone H3 lysine 27 acetylation (H3K27ac) overlapping with the proximal and distal HBZ-binding peaks [56] (Fig 3A). H3K27 is acetylated by p300/CBP [60,61], which forms a high-affinity interaction with HBZ [37,38]. Given this information, we compared coactivator-enrichment in HeLa cells expressing HBZ versus cells carrying the empty expression vector. Consistent with the enrichment of HBZ at the proximal peak in these cells, ChIP assays revealed that both p300 and CBP are enriched at this site in the presence of HBZ (Fig 4A and 4B, respectively). These results suggest that HBZ recruits p300/CBP to the *MYOF* gene.

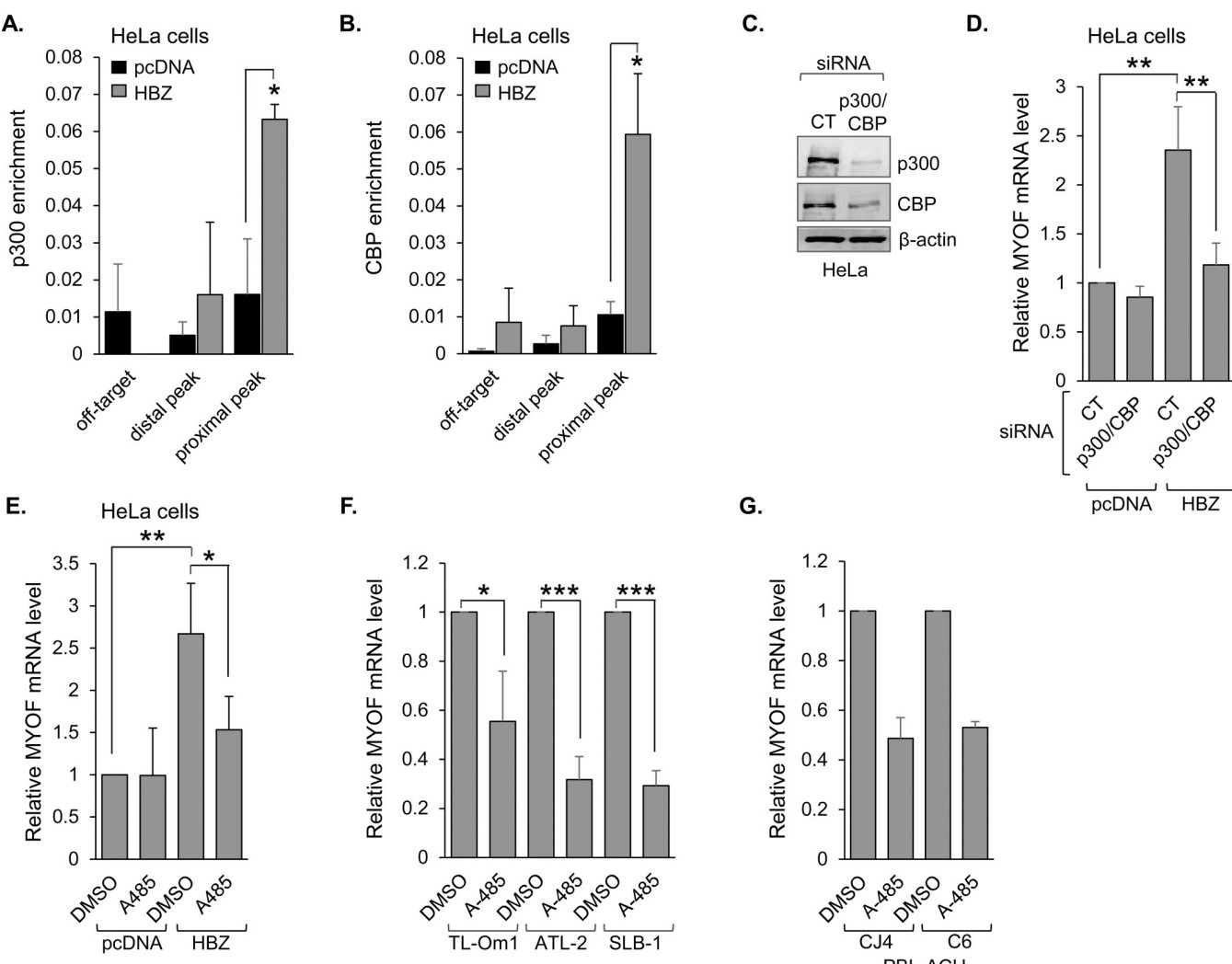

**Fig 4. HBZ activates *MYOF* transcription through recruitment of p300/CBP.** p300 (**A**) and CBP (**B**) bind the proximal *MYOF* peak in HeLa cells expressing HBZ. Graphs show average values from three independent ChIP assays using empty vector (pcDNA) and HBZ-expressing HeLa cells. Shown are levels of p300- and CBP-enrichment at the off-target site and at the distal and proximal peaks; * $p<0.05$. (**C**) siRNA-mediated depletion of p300 and CBP. HeLa cells were transfected with siRNAs targeting both p300 and CBP (p300/CBP) or an off-target siRNA (CT). Whole cell extracts (50 μg) were analyzed by Western blot using antibodies against p300, CBP and β-actin. (**D**) Depletion of p300 and CBP reduces HBZ-mediated *MYOF* expression. The graph shows qRT-PCR results averaged from four independent experiments with values normalized to those for the empty-vector clone (pcDNA) transfected with siRNA CT (set to 1); ** $p<0.01$. (**E**) Inhibition of p300/CBP KAT activity abrogates HBZ-mediated activation of *MYOF* transcription. HeLa cells expressing HBZ or carrying the empty vector (pcDNA) were treated with A485 (10 μM) or the carrier (DMSO) for 3h. The graph shows qRT-PCR results averaged from four independent experiments with values normalized to those for the empty-vector clone (pcDNA) treated with DMSO (set to 1); * $p<0.05$, ** $p<0.01$. Inhibition of p300/CBP KAT activity reduces *MYOF* transcription in (**F**) HTLV-1 infected T-cell lines (TL-Om1, ATL-2 and SLB-1) and (**G**) HTLV-1-immortalized primary human T-cell clones (CJ4 and C6). The indicated cell lines/clones were treated with A485 (10 μM) or the carrier (DMSO) for 3h. Graphs show qRT-PCR results averaged from three (F) and two (G) independent experiments with A485 values normalized to those for DMSO (set to 1) for each cell line/clone; * $p<0.05$, *** $p<0.001$.

To test the role of p300/CBP in transcriptional activation of *MYOF* by HBZ, we treated the empty vector and HBZ-expressing HeLa cells with an siRNA cocktail targeting both p300 and CBP (Fig 4C). Knockdown of the coactivators significantly reduced *MYOF* mRNA levels in the HBZ-expressing cells while producing no significant effect on transcript levels in the empty vector cells (Fig 4D). To gauge the contribution of p300/CBP KAT activity to transcriptional activation, we treated the HBZ-expressing and empty vector HeLa cells with A-485. This compound is a specific inhibitor of p300/CBP KAT activity [62]. Paralleling knockdown of the

coactivators, A-485 treatment significantly reduced *MYOF* mRNA levels in the HBZ-expressing cells without affecting transcript levels in the empty vector cells (Fig 4E). A-485 treatment also reduced *MYOF* mRNA levels in HTLV-1-infected T-cell lines as well as recently immortalized HTLV-1-infected T-cell clones (Fig 4F and 4G, respectively). Together, these results indicate that HBZ activates *MYOF* transcription through recruitment of p300/CBP.

## MyoF enhances HTLV-1 infection

For enveloped viruses, endosomal trafficking is an important process for incorporation of viral envelope and/or spike proteins into assembling virions [49,50]. Vesicle trafficking is also likely to be involved in infection through virological synapses [63], a mechanism used by HTLV-1 [12]. These premises led us to test whether MyoF contributes to HTLV-1 infectivity. We used a recently designed inhibitor of MyoF, WJ460, which disrupts the normal association of MyoF with intracellular vesicles [41]. Prior to analyzing viral infectivity, we determined that 24h-exposure of HTLV-1-infected T-cells to 0.2–1μM of WJ460 was not cytotoxic (S5 Fig). As T-cell infection by HTLV-1 predominantly occurs through direct cell-to-cell contact [12], we co-cultured DMSO (vehicle control)- and WJ460-treated HTLV-1-infected T-cells with target, reporter cells (Fig 5A). The two reporter cell lines used were CHOK1-Luc and Jurkat pmin-LUC-vCRE, which both contain promoters driving luciferase gene transcription that are *trans*-activated by the HTLV-1 Tax protein (only expressed following integration of the transferred virions; [31,64]). For Jurkat pminLUC-vCRE reporter cells, a 24h co-culture time was in the range of peak luciferase activity (S6A Fig). In assays using CHOK1-Luc target cells and ATL-2 or SLB-1 effector cells, treatment with WJ460 led to a significant reduction in reporter cell luciferase activity compared to treatment with DMSO (Figs 5B and S6B, respectively). Substituting CHOK1-Luc target cells with Jurkat pminLUC-vCRE T-cells also resulted in significantly reduced luciferase activity when co-cultured with WJ460-treated SLB-1 or ATL-2 effector cells (Figs 5C and S6C, respectively). Importantly, HTLV-1-infected C8166/45 cells, which express Tax but do not produce virions [65], did not activate luciferase transcription in target cells (Fig 5D), corroborating that luciferase activity was induced by HTLV-1 infection. In addition to analyzing these long-established HTLV-1-infected T-cell lines as effector cells, we tested primary human T-cells recently immortalized by HTLV-1 [66]. With these recently established effector cells, treatment of WJ460 also led to a significant reduction in luciferase activity when co-cultured with Jurkat pminLUC-vCRE cells (Figs 5E and S6D).

To verify that the reduction in target cell luciferase activity was due to inhibition of MyoF, we established stable expression of an shRNA targeting the *MYOF* transcript (shMYOF) or, as a control, an shRNA targeting GFP (shGFP) in both SLB-1 and ATL-2 cells (Fig 5F). Consistent with WJ460 treatment, knockdown of MyoF in SLB-1 cells led to a significant reduction in luciferase activity in both the CHOK1-Luc and Jurkat pminLUC-vCRE target cells (Fig 5G and 5H). Similarly, knockdown of MyoF in ATL-2 cells reduced luciferase activity in Jurkat-pminLUC-vCRE and CHOK1-Luc target cells (Figs 5I and S6E Fig). For these assays, we verified that shMYOF and shGFP cells did not exhibit significant differences in proliferation within the co-culture time period, and similar results were obtained when effector cells were irradiated to impair proliferation prior to co-culture (S6F and S6G Fig). Overall, these data indicated that MyoF provides a positive contribution to HTLV-1 infection.

## MyoF regulates the intracellular abundance of Env

The role of MyoF in endosomal recycling and concomitant regulation of receptor stability [42,45,67,68] implied it might affect the stability and intracellular trafficking of certain HTLV-1 proteins that compose the virion. We began to examine this possibility first using WJ460 to inhibit

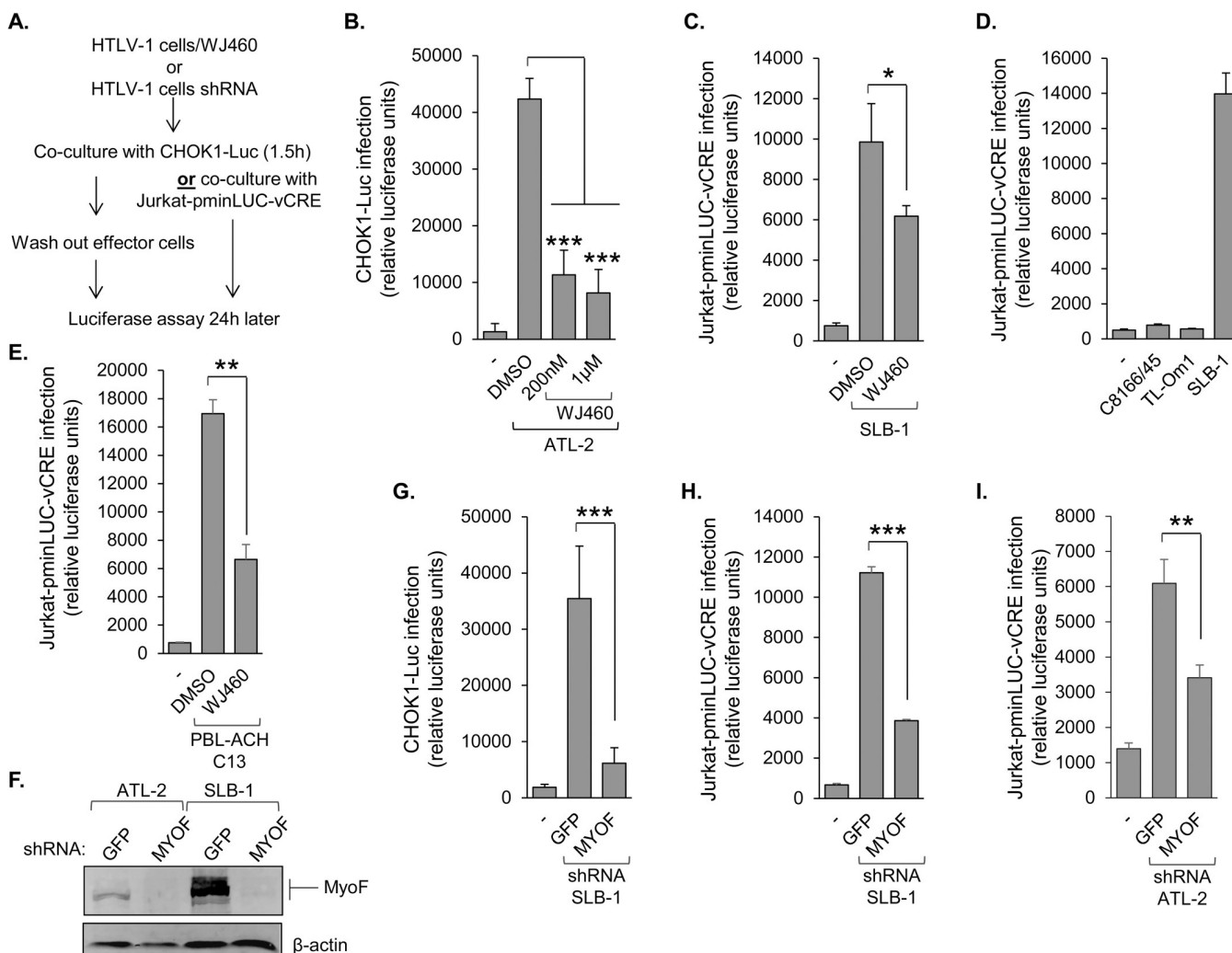

**Fig 5. MyoF enhances HTLV-1 infection. (A)** The flow diagram shows the co-culture/infection assay procedure using CHOK1-Luc or Jurkat-pminLUC-vCRE cells as target cells. **(B)** Inhibition of MyoF reduces HTLV-1 infection. ATL-2 cells were treated with DMSO or WJ460 (200nM or 1μM) prior to co-culture with CHOK1-Luc cells. **(C)** SLB-1 cells were treated with DMSO or 1 μM WJ460 prior to co-culture with Jurkat-pminLUC-vCRE cells. **(D)** C8166/45 and SLB-1 cells (both high Tax expression) and TL-Om1 cells (no Tax expression) were co-cultured with Jurkat-pminLUC-vCRE cells. **(E)** An HTLV-1-immortalized primary human T-cell clone (C13) was treated with DMSO or 1 μM WJ460 prior to co-culture with Jurkat-pminLUC-vCRE cells. Graphs (B), (C) and (D) show luciferase assay results averaged from at least three replicates for each condition of a single experiment and are representative of three independent experiments; * p<0.05, *** p<0.001. Graph (E) shows luciferase assay results average from three replicates for each condition of a single experiment and is representative of two independent experiments; ** p<0.01. **(F)** Knockdown of MyoF expression reduces HTLV-1 infection. MyoF expression in control (GFP) and MyoF knockdown ATL-2 and SLB-1 cells. Whole cell extracts (50 μg) were analyzed by Western blot using antibodies against MyoF and β-actin. **(G)** SLB-1 cells stably expressing an shRNA targeting GFP (negative control) or *MYOF* mRNA were co-cultured with CHOK1-Luc cells. **(H)** SLB-1 cells stably expressing an shRNA targeting GFP (negative control) or *MYOF* mRNA were co-cultured with Jurkat-pminLUC-vCRE cells. **(I)** ATL-2 cells stably expressing an shRNA targeting GFP (negative control) or *MYOF* mRNA were co-cultured with Jurkat-pminLUC-vCRE cells. Graphs show luciferase assay results averaged from at least three replicates for each infection condition of a single experiment and is representative of at least three independent experiments; ** p<0.01, *** p<0.001.

MyoF. In ATL-2 cells treated with WJ460, the level of SU was reduced, while levels of the other viral proteins examined remained relatively constant, including that of the Env precursor (Pr), gp62 (Fig 6A and 6C). The effect of WJ460 on SU was recapitulated in SLB-1 cells (Fig 6B and 6C). The observation that WJ460 treatment did not increase the level of gp62 suggested that the decrease in SU was not due to inhibition of furin-mediated cleavage of the precursor. Corroborating these results, knockdown of MyoF also led to a reduction in SU (Fig 6D and 6E).

To further analyze the effect of MyoF on Env abundance, we used HEK293T cells, which do not express MyoF. Transfecting these cells with an Env expression vector alone resulted in a weak western blot signal for SU that was substantially increased by co-transfection of a MyoF expression vector (Fig 6F). In these experiments, SU was enriched through binding to lectin-agarose beads, as ectopic expression of the mature form of Env (SU-TM) in uninfected cells is known to be low [26]. Consistent with results using the HTLV-1 T-cell lines, WJ460 reduced the level of SU in HEK293T cells co-transfected with the Env and MyoF expression vectors (Fig 6G). We also examined the effect of MyoF on TM abundance by inserting a Myc tag within its cytoplasmic domain as described [29], given that there is no available antibody to TM. Like SU, the level of TM was elevated in the presence of MyoF (Fig 6H). Together, these results support a role for MyoF in regulating the abundance of the mature form of Env.

## MyoF restricts lysosomal degradation of Env

MyoF and SU were not coimmunoprecipitated from whole cell extracts, suggesting that they do not interact. Therefore, we investigated the possibility that general effects of MyoF on the endosomal compartment influenced the intracellular trafficking of Env. To test this hypothesis, we first probed for changes in various markers of the endosomal compartment, as levels of some markers appear to be regulated by MyoF in other cell types [48,52,69]. We found that inhibition of MyoF with WJ460 did not significantly affect any of the markers examined (Fig 7A). However, stable shRNA-mediated knockdown of MyoF led to a reduction in the level of caveolin-1 (Cav1; Fig 7B). This observation is consistent with previous studies showing positive regulation of Cav1 levels by MyoF [48,69]. Cav1 is an essential structural protein of caveolae, which are plasma membrane invaginations that can undergo endocytosis [70]. While endocytosis of Env is believed to occur through a clathrin-dependent mechanism [27], it was possible that this process might separately involve a caveolin-dependent mechanism influenced by MyoF. To assess whether MyoF affects endocytosis of Env, we compared levels of SU at the surface of SLB-1 shMYOF and shGFP cells using flow cytometry. Surprisingly, we did not observe a significant difference in levels of surface SU (Fig 7C; a separate antibody that recognizes a distinct epitope of SU produced the same result). Similarly, we did not detect a difference in SU on the surface of DMSO- versus WJ460-teated SLB-1 cells. These results indicate that MyoF does not affect endocytosis of Env.

The cytoplasmic-exposed region of Env is devoid of lysine residues required for degradation through the ubiquitin-proteosome pathway, indicating that Env turnover occurs *via* lysosomal degradation. Indeed, retroviral Env proteins have been found in lysosomes [71,72]. These observations led us to hypothesize that MyoF regulates Env abundance by limiting Env trafficking to lysosomes. Analogous mechanisms have been reported for MyoF-dependent trafficking of Dishevelled-2 and Insulin Like Growth Factor 1 Receptor [43,45]. To test our hypothesis, we supplemented WJ460-treated SLB-1 cells with the lysosomal protease inhibitors, leupeptin and E-64, and found that the protease inhibitors partially restored the level of SU (Fig 7D). Similar results were obtained using ATL-2 cells (S7 Fig). Consistent with these data, Env exhibited greater overlap with the lysosomal marker, LAMP-2, in SLB-1 shMYOF cells than in shGFP cells (Fig 7E). While the antibody used recognizes both gp62 and SU, the Env associated with LAMP-2 is expected to primarily represent SU. This premise is consistent with the absence of a significant effect of MyoF knockdown or functional inhibition on levels of the gp62 precursor. Therefore, these overall results indicate MyoF reduces trafficking of SU to lysosomes for degradation.

We then analyzed effects of inhibitors of late endosomal trafficking on viral infection. SLB-1 cells were treated with individual inhibitors, washed and then co-cultured with CHOK1-Luc

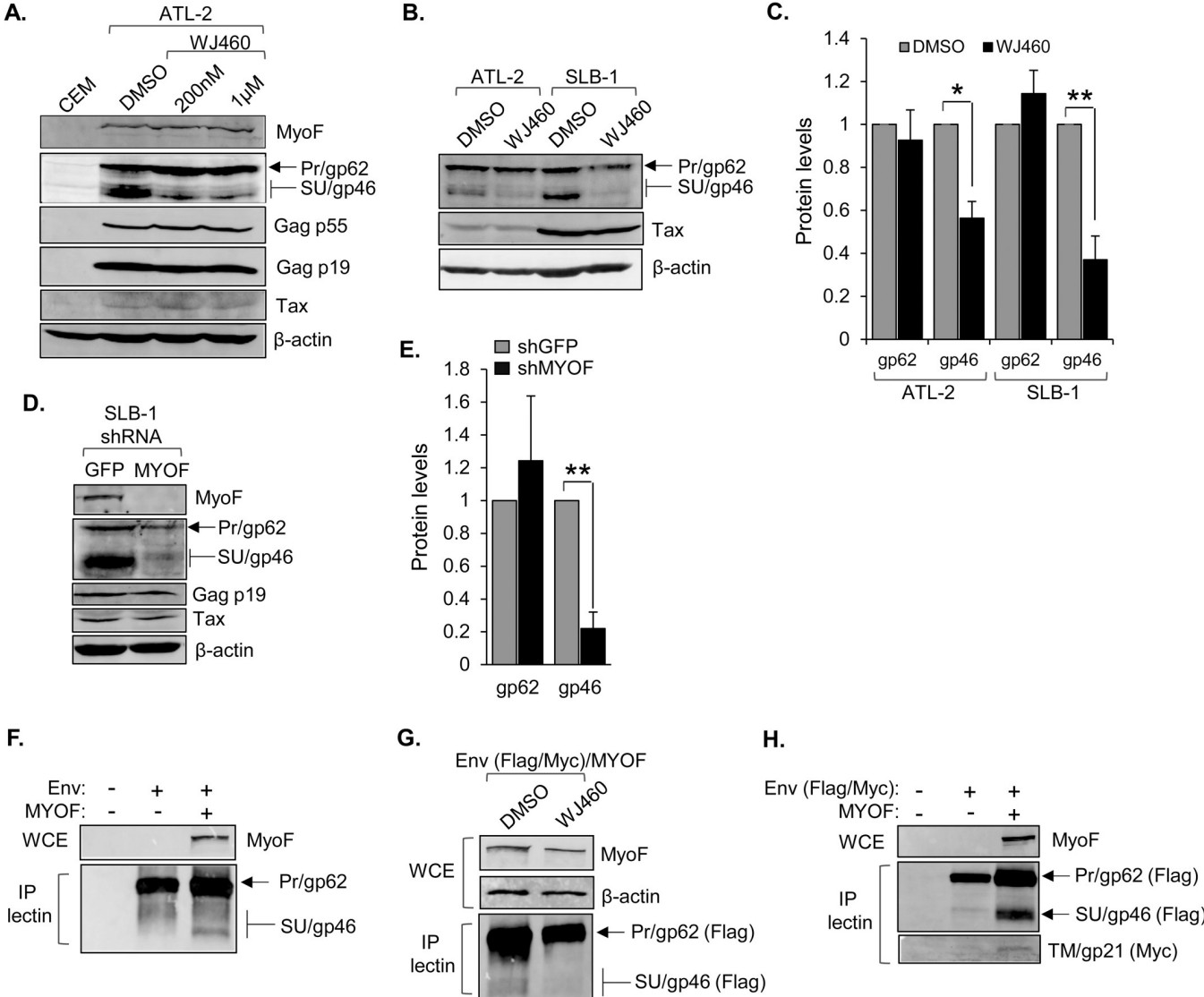

**Fig 6. MyoF regulates the intracellular abundance of HTLV-1 Env. (A)** Inhibition of MyoF reduces levels of SU (gp46). ATL-2 cells were treated with DMSO or WJ460 (200 nM or 1 µM) for 24h. Whole cell extracts (50 µg) from uninfected CEM and the treated cells were analyzed by Western blot using antibodies against MyoF, β-actin and the viral proteins Tax, Gag p55, Gag p19, and SU/Pr. **(B)** ATL-2 and SLB-1 cells were treated with DMSO or 1 µM WJ460 for 24h. Whole cell extracts (100 µg for Tax; 50 µg for the others) were analyzed by Western blot using antibodies against β-actin and the viral proteins Tax, and SU/Pr. **(C)** The graph shows quantification of band intensities of Pr (gp62) and SU (gp46) normalized to band intensities of β-actin averaged from three and four independent experiments for ATL-2 and SLB-1 cells, respectively. Error bars show standard deviations; [*] $p<0.05$, [**] $p<0.01$. **(D)** Knockdown of MyoF expression reduces levels of SU (gp46). Whole cell extracts were prepared from SLB-1 cells stably expressing an shRNA targeting GFP (negative control) or *MYOF* mRNA, and 50 µg from each cell line was analyzed by Western blot using antibodies against MyoF, β-actin and the viral proteins Tax, Gag p19, and SU/Pr. **(E)** The graph shows quantification of band intensities of Pr (gp62) and SU (gp46) normalized to band intensities of β-actin averaged from three independent experiments. Error bars show standard deviations; [**] $p<0.01$. **(F)** Ectopic co-expression of MyoF with Env in HEK293T cells increases SU (gp46) abundance. Cells (1 x 10⁶) were transfected with pcDNA-GFP-HA-MyoF (3 µg) and pCMV-HTLV-1-Env (1 µg). Env was co-precipitated with Lens Culinaris Agglutinin lectin bound to agarose beads (P-lectin) from 879 µg of whole cell extracts (WCE) and analyzed along with 50 µg WCE by Western blot using antibodies against MyoF, and SU/Pr. **(G)** Inhibition of MyoF ectopically co-expressed with Env in HEK293T cells decreases SU/gp46 abundance. Cells (1 x 10⁶) were transfected with pcDNA-GFP-HA-MyoF (3 µg) and pCMV-HTLV-1-Env-Flag-Myc (3 µg) expression vectors and treated with DMSO or WJ460 (1 µM) for 24h. Env was co-precipitated with Lens Culinaris Agglutinin lectin bound to agarose beads (P-lectin) from 768 µg of whole cell extracts (WCE) and analyzed along with 50 µg WCE by Western blot using antibodies against MyoF, β-actin and the Flag-epitope tag of Env. **(H)** Ectopic co-expression of MyoF with Env in HEK293T cells increases TM (gp21) abundance. Cells (1 x 10⁶) were transfected with pcDNA-GFP-HA-MyoF (3 µg) and pCMV-HTLV-1-Env-Flag-Myc (3 µg). Env was co-precipitated with Lens Culinaris Agglutinin lectin bound to agarose beads (P-lectin) from 1190 µg of whole cell extracts (WCE) and analyzed along with 50 µg WCE by Western blot using antibodies against MyoF, and the Flag- and Myc-epitope tags of Env.

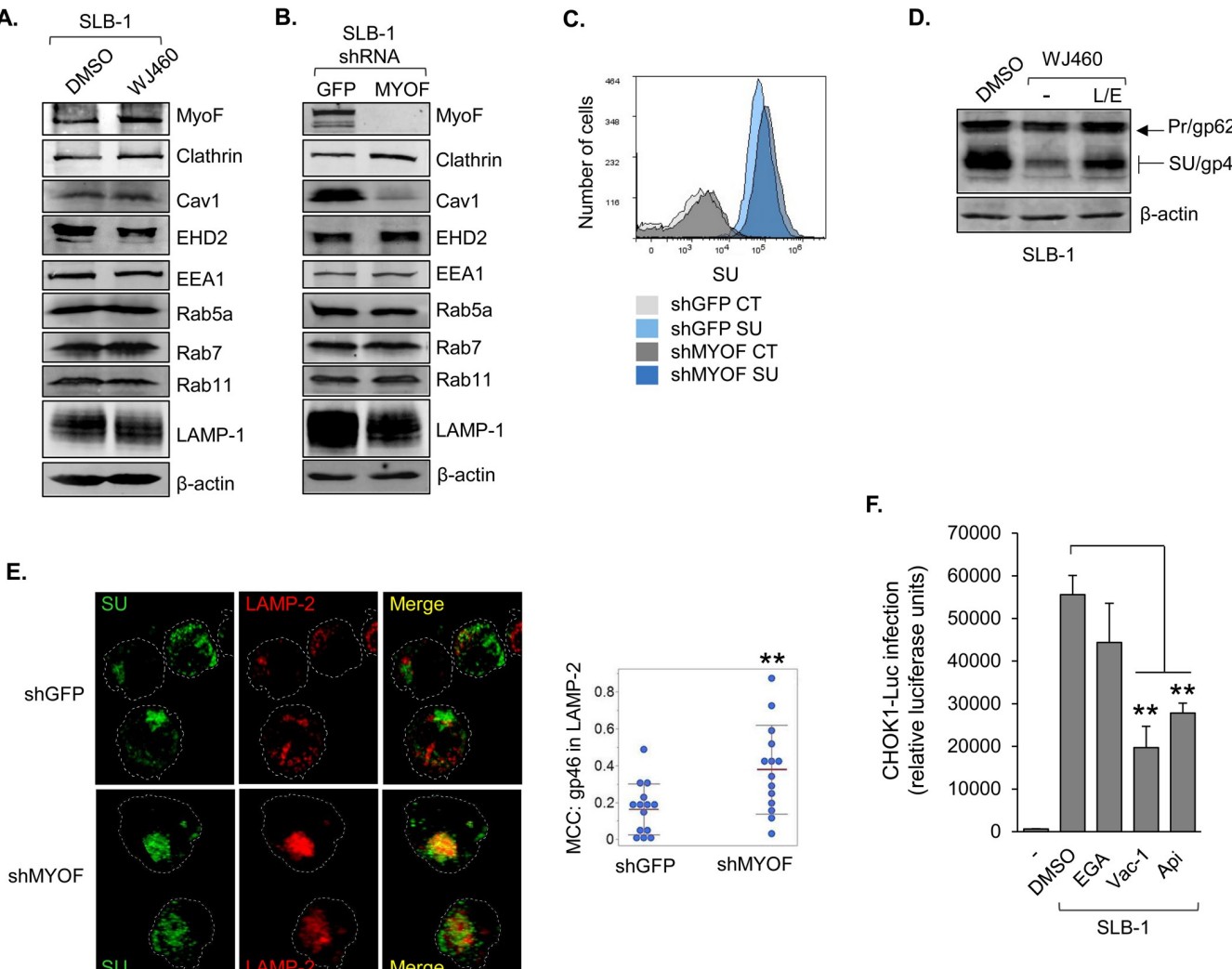

**Fig 7. MyoF restricts lysosomal degradation of Env. (A)** Levels of endosomal proteins in HTLV-1 infected cells following WJ460 treatment. SLB-1 cells were treated with DMSO or 1 μM WJ460 for 24 h. Whole cell extracts (25 to 50 μg) were analyzed by Western blot using antibodies against the indicated proteins. **(B)** Knockdown of MyoF expression reduces the level of caveolin-1 (Cav1). Whole cell extracts (25 to 50 μg) were prepared from SLB-1 cells stably expressing an shRNA targeting GFP (negative control) or *MYOF* mRNA and analyzed by Western blot using antibodies against the indicated proteins. **(C)** Knockdown of MyoF expression does not affect the level of SU (gp46) at the cell-surface. SLB-1 cells stably expressing an shRNA targeting GFP (negative control) or *MYOF* mRNA were labeled with a primary antibody against SU/Pr followed by an APC-conjugated secondary antibody, fixed and analyzed by flow cytometry. Histograms are representative of two independent experiments and show relative cell surface labeling as follows: secondary antibody alone (CT) and SU-labeled shGFP cells, light gray and light blue, respectively; secondary antibody alone (CT) and SU-labeled shMYOF cells, dark gray and dark blue, respectively. **(D)** Inhibition of lysosomal proteases partially restores intracellular levels of SU (gp46) in HTLV-1 infected cells treated with WJ460. SLB-1 cells were treated with DMSO or 1 μM WJ460 with or without 5 mg/mL leupeptin (L) and 25 μM E64D (E) for 24 h. Whole cell extracts (50 μg) were analyzed by Western blot using antibodies against β-actin and SU/Pr. **(E)** Knockdown of MyoF expression increases SU (gp46) association with lysosomes. SLB-1 cells stably expressing an shRNA targeting GFP (negative control) or *MYOF* mRNA were fixed, permeabilized, labelled with Alexa Fluor 488- and Alexa Fluor 594-conjugated antibodies against SU/Pr and LAMP-2, respectively, and analyzed by confocal microscopy. The images show 3D projections constructed from z-stacks comprised of 0.57 μm optical slices. The graph shows Manders' Colocalization Coefficients (MCC) for the fraction of SU overlapping LAMP-2; ** $p<0.01$. **(F)** Inhibition of endosomal trafficking reduces HTLV-1 infection. SLB-1 cells were treated with DMSO, 10 μM EGA, 10 μM vacuolin-1 (Vac-1) or 1 μM apilimod (Api) for 4 h prior to co-culture with CHOK1-Luc cells. The graph shows luciferase assay results averaged from three replicates for each condition of a single experiment and are representative of three independent experiments; ** $p<0.01$.

target cells. The inhibitor, EGA, for which the cellular target is currently unknown, affects trafficking encompassing low-pH vesicles, such as endolysomes and lysosomes [73]. The inhibitors, apilimod and vacuolin-1, target the phosphoinositide kinase, PIKfyve [74,75], and

produce broader effects on endosomal trafficking that extend to trafficking encompassing multivesicular bodies (MVBs) [76,77]. While known to direct cargo to lysosomes, MVBs also direct cargo to the trans-Golgi network and the endocytic recycling compartment [78]. We found that SLB-1 cells treated with EGA did not exhibit a significant difference in infection compared to cells treated with DMSO; however, cells treated with vacuolin-1 or apilimod showed a significant reduction in infectivity (Fig 7F). These data indicate that endosomal trafficking through MVBs in effector cells is important for HTLV-1 infection.

### MyoF enhances HTLV-1-infected T-cell adhesion

HTLV-1 infection predominantly occurs through contact between effector T-cells and target cells, which can induce formation of a virological synapse through which virions are transferred [12]. Induction of the virological synapse requires ICAM-1 on the effector cell to bind LFA-1 on the target cell [13]. As MyoF is known to regulate receptor recycling, we tested whether it affected the abundance of ICAM-1 on the surface of HTLV-1-infected T-cells. Flow cytometric analysis of surface-labeled cells did not reveal a significant difference in ICAM-1 levels between SLB-1 shGFP and shMYOF cells (Fig 8A).

Separately, we examined whether MyoF contributed to the general adhesion properties of HTLV-1-infected T-cells. In these experiments, SLB-1 cells labeled with a fluorescent marker (calcein AM) were added onto adherent CHO or CHO-LFA-1 cells. The latter cells were

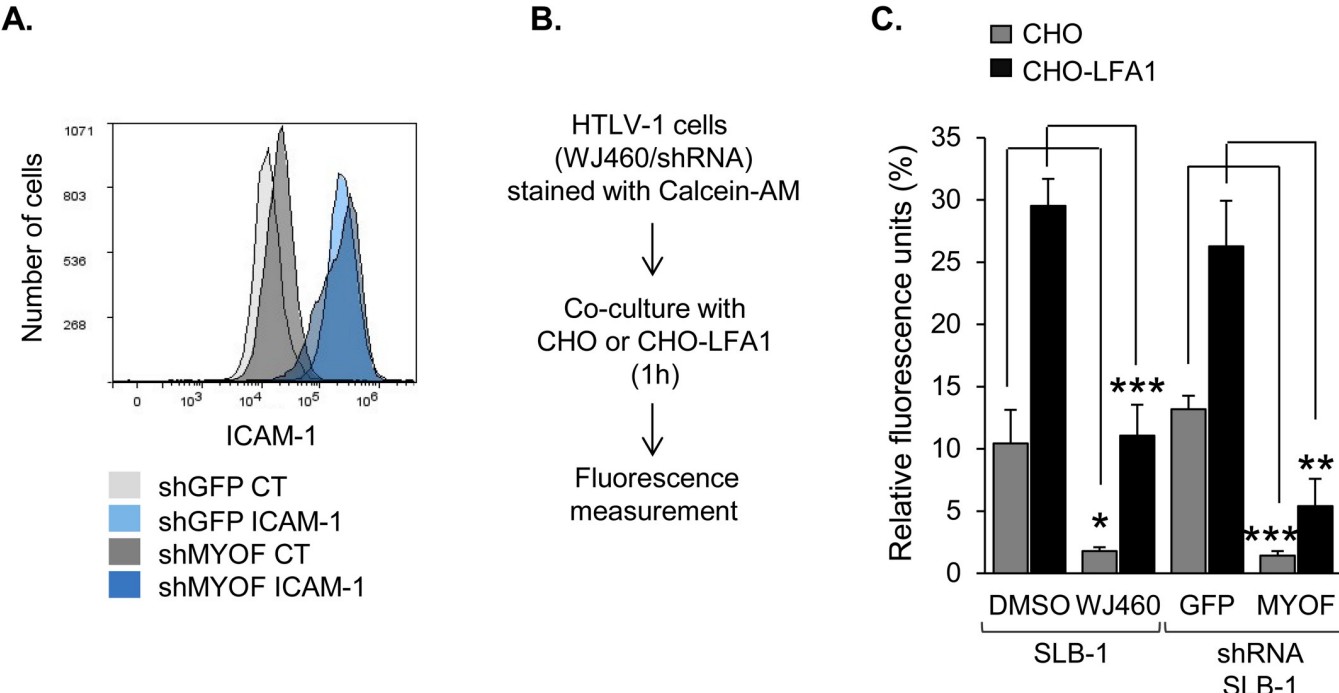

**Fig 8. MyoF enhances HTLV-1-infected T-cell adhesion without affecting cell surface abundance of ICAM-1. (A)** Knockdown of MyoF expression does not affect the level of ICAM-1 at the cell-surface. SLB-1 cells stably expressing an shRNA targeting GFP (negative control) or *MYOF* mRNA were labeled with an antibody against ICAM-1 followed by a FITC-conjugated secondary antibody, fixed and analyzed by flow cytometry. Histograms show relative cell surface labeling as follows: secondary antibody alone- (CT) and ICAM-1-labeled shGFP cells, light gray and light blue, respectively; secondary antibody alone- (CT) and ICAM-1-labeled shMYOF cells, dark gray and dark blue, respectively. **(B)** Knockdown or inhibition of MyoF reduces adhesion to CHO cells independent of LFA-1 expression. The flow diagram shows the co-culture/adhesion assay procedure using CHO or CHO-LFA-1 adherent cells. **(C)** Adhesion assays of HTLV-1 infected cells stably expressing shGFP or shMyoF to adherent cells. The graph shows binding of SLB-1 cells pretreated with DMSO or 1 μM WJ460, or SLB-1 cells stably expressing an shRNA targeting GFP (negative control) or *MYOF* mRNA, to CHO (grey) or CHO-LFA1 (black) cells. Values are the average of three replicates from one experiment and representative of three independent experiments; * *p*<0.05, ** *p*<0.01, *** *p*<0.001.

established by stable co-transfection of expression vectors for the integrins β2 (CD18) and αL (CD11a), which form the LFA-1 heterodimer [31]. Following co-culture, the unbound T-cells were removed by washing, and remaining cells were lysed and analyzed for fluorescence (Fig 8B). Surprisingly, we found that inhibition or knockdown of MyoF significantly impaired cell adhesion independent of LFA-1 expression on the target cells (Fig 8C). Together, these results indicated that MyoF plays a positive role in cell adhesion, but this function does not involve adhesion through ICAM-1/LFA-1.

## MyoF increases levels of extracellular and virion-associated Env

Given the effects of MyoF on Env trafficking and on the intracellular abundance of Env, we tested whether MyoF knockdown affected Env incorporation into assembling virions. Such an effect would be reflected by a change in the level of SU associated with cell-free virions produced following MyoF knockdown. However, prior to analyzing virions, we first tested for a difference in the overall abundance of SU in culture media from SLB-1 shGFP and shMYOF cells. Western blot analysis of TCA-precipitated proteins from filtered culture supernatants revealed less SU in the shMYOF precipitate than in the shGFP precipitate (Fig 9A). We then compared SU levels associated cell-free virions obtained by ultracentrifugation of filtered culture supernatants from the knockdown cells. Paralleling the general analysis of the culture media, the level of SU was lower in the shMYOF specimen while the level of Gag p19 remained relatively unchanged (Fig 9B). Quantification of Gag p19 by ELISA revealed similar levels of this protein in culture media from SLB-1 shGFP and shMYOF cells, and separately, in media

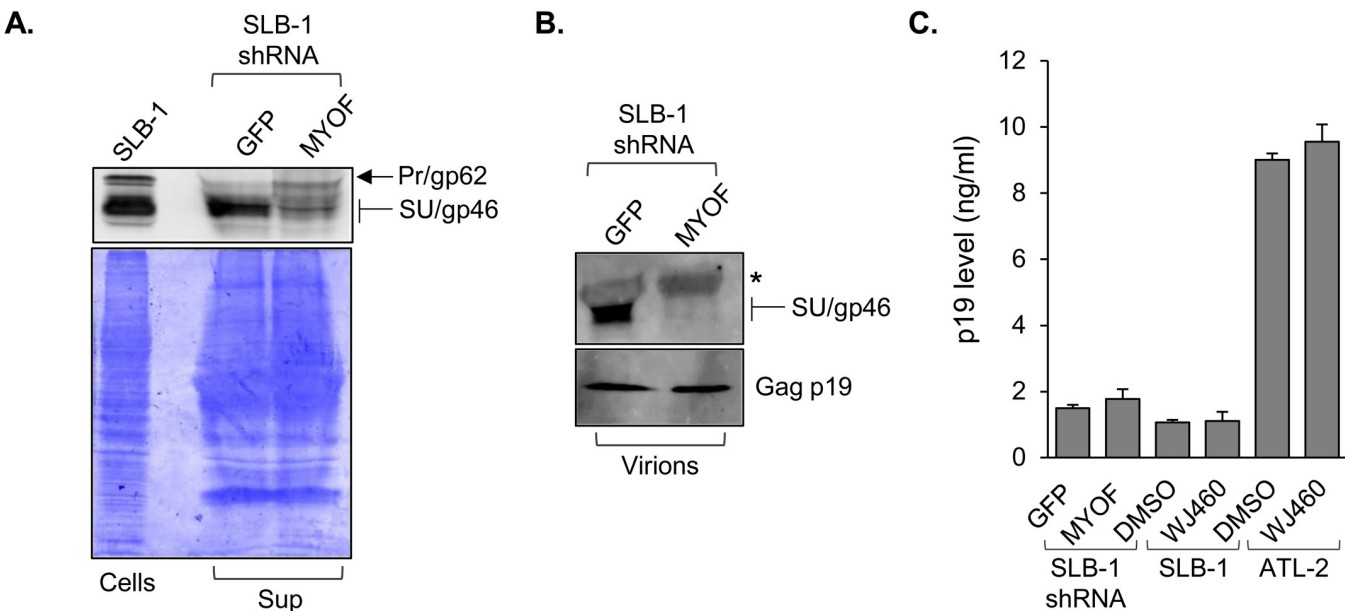

**Fig 9. MyoF increases levels of total extracellular and virion-associated SU (gp46). (A)** Knockdown of MyoF expression reduces overall levels of SU in the culture medium of HTLV-1 cells. TCA-precipitated proteins from culture media of SLB-1 cells stably transfected with shRNA targeting GFP or *MYOF* mRNA were divided into equal fractions and analyzed by Western blot using an antibody against SU/Pr (upper panel) and, separately, by Coomassie blue-staining following SDS-PAGE (lower panel). Whole cell extract from SLB-1 cells served as a positive control. **(B)** Knockdown of MyoF expression reduces levels of SU in cell-free HTLV-1 virions. Culture media from SLB-1 cells stably expressing an shRNA targeting GFP or *MYOF* mRNA were filtered, ultracentrifuged and analyzed by Western blot using antibodies against SU/gp46 and Gag p19; * denotes a nonspecific band from the serum. **(C)** Knockdown or inhibition of MyoF does not affect levels of virus released from the cells. Clarified culture media from SLB-1 cells stably expressing an shRNA targeting GFP or *MYOF* mRNA and SLB-1 and ATL-2 cells treated with DMSO or 1 μM WJ460 for 24 h were analyzed by ELISA to quantify levels of Gag p19. The graph shows values averaged from three replicates.

from SLB-1 cells treated with DMSO and WJ460 (Fig 9C). These results indicate that knockdown or inhibition of MyoF leads to a reduction in Env associated with free virus.

## Discussion

The viral protein, HBZ, functions as a transcription factor to regulate transcription of the HTLV-1 provirus as well as expression of cellular genes [32]. We previously reported that HBZ increases ICAM-1 expression in HTLV-1-infected cells to enhance the efficiency of infection [31]. ICAM-1 is central to the cell-contact-mediated mechanism of viral infection, as it promotes adhesion between effector and target cells and promotes formation of a virological synapse [13,18]. In the current study, we provide evidence that HBZ induces expression of MyoF, which also contributes to infection. This effect correlated with the role of MyoF in modulating endosomal trafficking, which affected the intracellular abundance of HTLV-1 Env and the level of Env associated with virus particles.

HBZ increased MyoF expression in HTLV-1-infected T-cells and in HeLa cells stably expressing HBZ. Our data indicate that HBZ activates *MYOF* transcription by binding to one or two sites (cell type-dependent) within the gene. Both HBZ-binding sites are located far from the transcriptional start site (TTS): one ~19 kb downstream (proximal peak) and the other ~103 kb downstream (distal peak) of the TSS of *MYOF* isoforms a and b. The DNA sequences of both peaks contain full or partial TPA responsive elements (TREs), which are recognized by AP-1 dimers and, in this study, correlate with identification of JunB and c-Jun as potential partners for HBZ [33,58]. Indeed, HBZ is known to form heterodimers with the Jun proteins, which also includes JunD, which was not significantly enriched within the HBZ-binding peaks [32]. It is possible that the anti-JunD antibody we used, while designated ChIP-compatible, was not suitable for use with the HTLV-1-infected T-cells or for our ChIP workflow.

Co-enrichment of c-Jun and JunB with HBZ on the *MYOF* gene is intriguing, as, so far, HBZ interactions with Jun proteins are generally thought to repress transcription from TREs [58]. Indeed, the basic region of HBZ lacks the classical amino acid motif responsible for TRE-binding. Exceptionally, complexes formed by HBZ and JunD can associate with Sp1 to activate transcription [59]. In this complex, Sp1 is bound to the DNA, and HBZ and JunD are bound through protein-protein interactions rather than interactions with the DNA [35,36]. Data from the UCSC Genome Browser (http://genome.ucsc.edu; [79]) did not show Sp1-enrichement or Sp1 binding sites (GC boxes) within the sequences of the HBZ-binding peaks, suggesting that activation of *MYOF* gene transcription does not occur through a complex composed of Sp1, a Jun member, and HBZ.

Unlike heterodimers formed between HBZ and Jun proteins, those formed between HBZ and small Mafs have been shown to bind DNA [33]. Indeed, we previously demonstrated that HBZ activates *HMOX1* transcription by forming complexes with the small Maf, MafG, on MAREs located within the *HMOX1* promoter [34]. Interesting, the core sequence of the MAREs corresponds to a TRE. Therefore, it is possible that specific permutations within the TRE along with specific sequences surrounding the TRE may favor binding of heterodimers composed of HBZ and one of the Jun proteins. Consistent with this hypothesis, a recent *in silico* analysis revealed that TRE-like sequences were the predominant (known) motifs found in HBZ-enriched regions across the genome of ATL cells [80]. Furthermore, HBZ was primarily enriched at distal enhancer regions, with most enrichment sites located from 10 to 100 Kb from a TSS [80], which are distances that coincide with HBZ enrichment within the *MYOF* gene. It would be interesting to determine whether heterodimers composed of HBZ and one of the Jun proteins are capable of binding specific DNA elements. In addressing this issue, the sequences corresponding to the HBZ-binding sites of the *MYOF* gene may be useful.

Our results indicate that recruitment of p300 and/or CBP to the *MYOF* gene by HBZ is essential for transcriptional activation. Given the constitutive expression of HBZ in HTLV-1-infected T-cells, we were limited to evaluating p300/CBP recruitment in HeLa cells by comparing the cells that stably expressing HBZ to those that carry the empty vector. As with HBZ, p300 and CBP were only enriched at the proximal peak in the HBZ-expressing HeLa cells. However, it is likely that HBZ recruits the coactivators to both the proximal and distal peaks in HTLV-1 infected T-cells. This premise is based on the observation that peaks of histone H3 K27 acetylation overlap with both HBZ-binding peaks. H3K27ac is specifically established by p300/CBP KAT activity and marks transcriptionally active enhancers [60,61]. In addition, to their KAT activity, p300 and CBP serve as scaffolds for the recruitment of other transcriptional regulators [39]. Of these two general properties of p300/CBP, the KAT activity appears to provide the primary contribution to activating *MYOF* transcription. Indeed, in HeLa cells, the selective inhibitor of p300/CBP KAT activity, A-485, almost fully abrogated the effect of HBZ on *MYOF* transcription, and in HTLV-1-infected T-cells, A-485 significantly reduced *MYOF* mRNA levels. Consistent with these observations, enhancer-based p300/CBP acetylate histones and transcriptional regulators, which is essential for formation of the RNA polymerase II (RNAPII) preinitiation complex and additionally to overcome promoter-proximal pausing of RNAPII [61,81].

In addition to *MYOF*, other cellular genes that affect the viral replication cycle or influence viral pathogenesis are likely activated by HBZ through recruitment of p300/CBP. To date, these genes include *BATF3*, which helps establish the gene expression profile of ATL cells [56], *CCR4*, which increases cell migration and is currently targeted in clinical trials to treat ATL patients [82], *DKK1*, which promotes ATL-associated hypercalcemia and formation of bone lesions [46,83], and *FOXP3*, which maintains the T-reg-like phenotype of HTLV-1 infected cells [84]. We believe that targeting the interaction between p300/CBP and HBZ would be an effective strategy to prevent the deregulated expression of these genes, thereby reducing the intercellular spread of HTLV-1 and pathogenic effects associated with infection.

In HTLV-1-infected T-cells, functional inhibition or knockdown of MyoF reduced infectivity. This effect was correlated with a decrease in levels of intracellular Env and Env associated with virus particles, but surprisingly, did not influence the level of Env at the cell surface. It is possible that Env detected at the surface of intact cells by flow cytometry mainly comprised virus-free Env rather than Env incorporated into virus particles. Indeed, for HTLV-1, virus particles remain associated with cells in an extracellular protein/carbohydrate matrix [15], which may obstruct antibody-binding [85]. The virus-free Env at the cell surface corresponds to homotrimers of the mature form of the protein. This pool of Env undergoes rapid endocytosis [28], which may represent an immune evasion tactic or a mechanism to avoid syncytia formation caused by the highly fusogenic capacity of HTLV-1 Env. Like most retroviruses, the cytoplasmic portion of the transmembrane subunit (TM) of HTLV-1 Env contains a YXXΦ motif (amino acids 479–482) involved in trafficking and endocytosis [27,86]. Φ denotes an amino acid with a bulk hydrophobic side chain, which for HTLV-1 Env, is an isoleucine. This motif, along with $Y_{476}$, mediates binding to adaptor protein 2 for clathrin-dependent endocytosis [27]. While MyoF plays a role in endocytosis, it is primarily involved in caveolin-dependent processes [48]. Therefore, MyoF is not expected to have a profound effect on the steady-state level of virus-free Env at the cell surface.

MyoF is also not expected to affect the initial processing of Env. HTLV-1 *env* mRNA is translated into the gp62 polypeptide precursor that undergoes proteolytic cleavage to produce SU and TM during trafficking through the *trans*-Golgi network [19]. Any effect of MyoF on Env maturation should be reflected by a change in the level of gp62 when MyoF is functionally inhibited or knocked down, which was not the case in this study.

Our results indicate that MyoF limits lysosomal degradation of Env, but not through an interaction with the viral protein. We hypothesize that MyoF regulates Env trafficking through its general effects on the endosomal compartment. MyoF contains a short C-terminal trans-membrane domain for insertion into intracellular membranes and a large cytoplasmic region harboring seven C2 domains designated C2A to C2G (ordered by N- to C-terminal position-ing) among other domains [40]. C2 domains are found in a wide variety of proteins, including other members of the ferlin family, and are generally involved in phospholipid- and/or pro-tein-binding [87]. Relevant to this study, the C2B domain of MyoF interacts with Epsin15 homology domain protein 1 and 2 (EHD1 and EHD2) [42,88]. These proteins function in intracellular membrane remodeling, playing an important role in the scission of vesicles from membrane networks that is required for cargo trafficking [89]. Interestingly, slow recycling of receptors, which occurs through the endocytic recycling compartment, is impeded by either knockdown of MyoF or, separately, deletion of the MyoF-binding domain in EHD1 [42,90]. One possible explanation for the effect of MyoF on Env stability is that, following endocytosis and incorporation into the early endosome, Env traffics to the endocytic recycling compart-ment (ERC). From the ERC Env then undergoes slow recycling to sites of viral assembly. Con-sidering this process, loss or functional inhibition of MyoF would impair transit of Env to the endosomal recycling compartment or impair slow recycling of Env from this compartment. Such effects could cause Env to be redirected to lysosomes for degradation.

In summary, we propose a tentative model diagraming how HBZ enhances HTLV-1 infec-tion by activating MyoF expression (Fig 10). We speculate that when its expression is activated by HBZ, MyoF diverts Env away from the lysosomal degradation pathway to the slow endoso-mal recycling pathway, which might facilitate incorporation of Env into assembling virions. Further work is needed to test this model.

## Materials and methods

### Plasmids

pMDLg/pRRE, pRSV-Rev and pMD2.G were gifts from Didier Trono (Addgene plasmid # 12251, 12253, 12259) [91]. Mission shRNA plasmids targeting GFP (SHC202) and *MYOF* (TRCN0000320396) were from MilliporeSigma. pCMV-HTLV-1-Env and pSG-Tax-His have been described [92,93]. pcDNA-GFP-HA-MyoF was a gift from Rikinari Hanayama [94]. pCMV-HTLV-1-Env-Flag-Myc was constructed by inserting the Flag sequence between serine 25 and cysteine 26 at the PvuII site in pCMV-HTLV-1-Env, and the Myc sequence between the YXXΦ and PDZ-binding motifs as described [29] using a synthetized fragment (GeneArt Gene Synthesis, Thermo Fisher Scientific) containing the Myc sequence that replaced the NsiI and SacII fragment. pLJM1-EGFP was a gift from David Sabatini (Addgene plasmid # 19319) [95]. To make pLJM1-EGFP-HBZ-Myc-His, the HBZ-Myc-His sequence from pcDNA3.1--Myc-His HBZ SP1 [96] was cloned into the EcoRI site of pLJM1-EGFP using the Gibson Assembly Cloning Kit (New England BioLabs).

### Cell culture and generation of cell lines

Jurkat, Jurkat pminLUC-viral CRE [31], CEM, MOLT-4, Hut-78, ATL-2s, C10/MJ, MT-2, MT-4, and SLB-1 cells were cultured in Iscove's modified Dulbecco medium (IMDM). Pri-mary CD4$^+$ lymphocytes, HTLV-1-immortalized lymphocyte clones, HSB-2, Sup-T1, Myla, 1185, MT-1, SP, Hut-102, TL-Om1 and BxPC3 cells were cultured in Roswell Park Memorial Institute (RPMI) medium. HeLa clones [46], HEK293T, CHO, CHO-LFA-1 [31] and CHOK1-Luc [64] cells were cultured in Dulbecco's modified Eagle's medium (DMEM). All cells were supplemented with 10% FBS or 10% Fetalplex animal serum (Gemini Bio-Products)

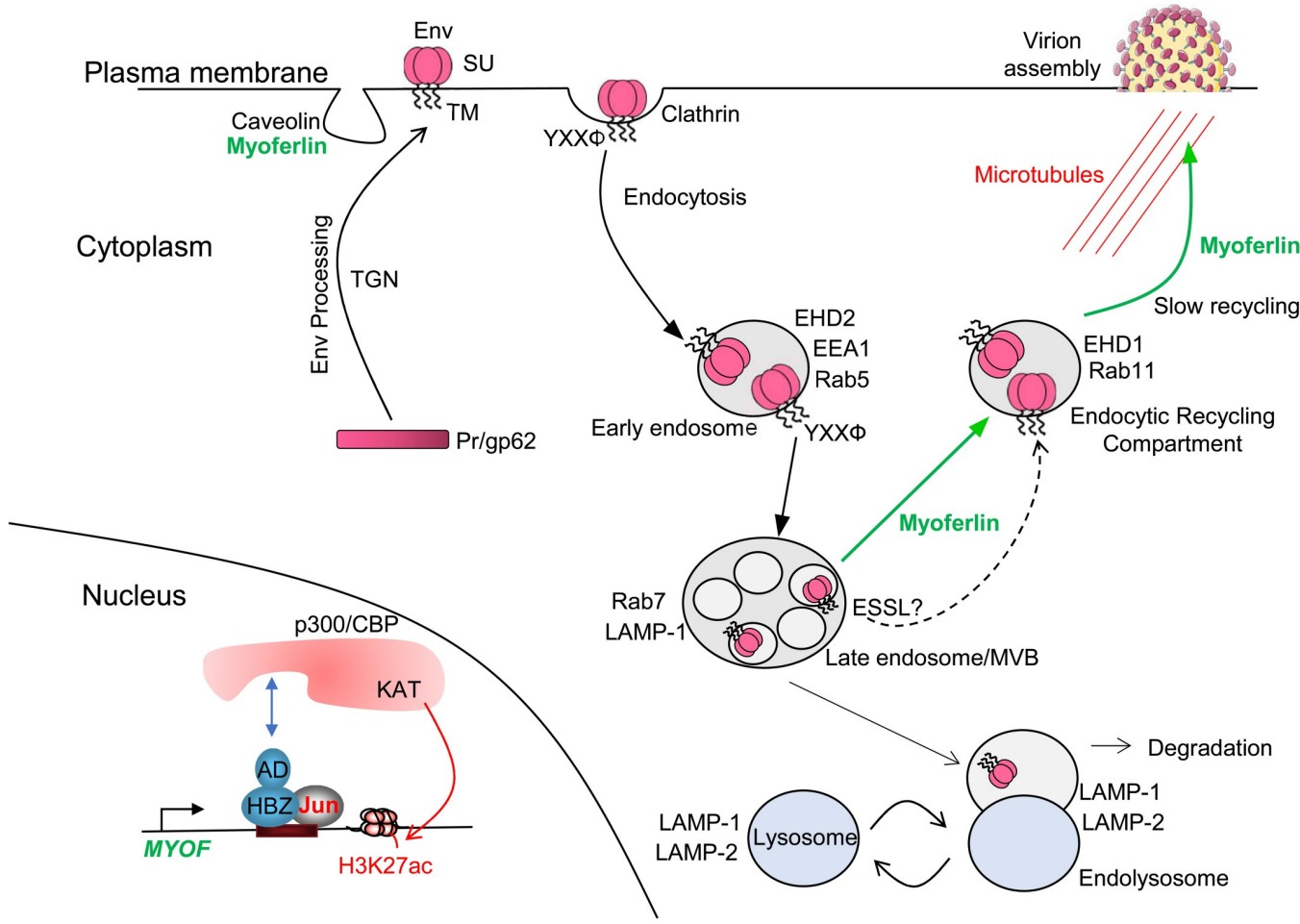

**Fig 10. Proposed model of MyoF-mediated Env trafficking in HTLV-1-infected T-cells.** Through its activation domain (AD), HBZ recruits p300/CBP to the *MYOF* gene enhancers to activate transcription. Env is processed to its mature form through the Trans-Golgi Network (TGN) and incorporated into the plasma membrane. The YXXΦ motif in Env mediates clathrin-dependent endocytosis and trafficking of Env to lysosomes. However, the presence of MyoF causes Env to be redirected to the endosomal recycling compartment from which Env traffics to sites of virion assembly. Apart from MyoF the ESSL domain separately reduces lysosomal-mediated degradation of Env. One component of this figure was drawn using a picture from Servier Medical Art, which are licensed under a Creative Commons Attribution 3.0 Unported License (https://smart.servier.com).

and 2 mM l-glutamine, 100 U/ml penicillin and 50 μg/ml streptomycin. Jurkat pminLUC-viral CRE cells were supplemented with 1.5 mg/ml G418. HeLa clones, CHO-LFA-1 and CHOK1-Luc cells were supplemented with 0.5 mg/ml G418. Primary lymphocytes, lymphocyte clones, SP, 1185 and Hut-102 cells were cultured with IL-2. HTLV-1-immortalized lymphocyte clones were established as described [66]. Primary human CD4[+] lymphocytes were obtained from human buffy coat or leukocyte reduction system cones (ZenBio or STEMCELL Technologies, respectively) and purified by centrifugation through Ficoll-Paque PLUS medium (GE Healthcare) and then using the CD4[+] T Cell Isolation Kit (STEMCELL Technologies) according to the manufacturers' instructions. A fraction of the isolated CD4+ lymphocytes was activated in culture wells coated with anti-CD3 and anti-CD28 antibodies. SLB-1 and ATL-2 shGFP and shMYOF cells were established by lentivirus transduction as follows. HEK293T cells (5 x 10[6]) were plated in 10 cm dishes, cultured overnight and then transfected with 17 μg pMDLg/pRRE, 8 μg pRSV-Rev, 11 μg of pMD2.G and 34 μg shRNA MISSION plasmid (Millipore-Sigma) using calcium phosphate. The culture medium was replaced 24h post-transfection with

6 mL supplemented IMDM, and the cells were cultured for another 24h. The culture medium (viral supernatant) was then passed through a 0.4 µm polyethersulfone (PES) filter; the transfected HEK293T cells were maintained in culture by adding back 6 mL fresh supplemented IMDM to expose cells to a second round of virus. Pelleted SLB-1 and ATL-2 cells ($3 \times 10^6$) were resuspended in the filtered viral supernatant with 12 ug/mL polybrene. The transduction processes were repeated. Cells were cultured for 48h and then supplemented with 6 µg/mL (SLB-1 cells) or 0.5 µg/mL (ATL-2 cells) of puromycin. Transient expression of HBZ-6x His in SLB-1 cells was achieved by a similar transduction method but using 44 µg of pLJM1-EGFP-HBZ in place of the shRNA plasmid.

### RNA extraction, cDNA synthesis, and quantitative real-time PCR

RNA was extracted from cells using the TRIzol Reagent (Thermo Fisher Scientific), and cDNA was synthesized using the Revert Aid kit (Thermo Fisher Scientific) or IScript (BioRad) as described by the manufacturers. Random hexamer primers were used for *MYOF* cDNA synthesis; oligo(dT) primers were inefficient for *MYOF* due to the 5' proximal position of the amplicon far from the poly-A tail of the mRNA. Except for copy number determination, oligo (dT) primers were used for *hbz* cDNA synthesis to try to reflect *hbz* mRNA fated for translation rather than the portion retained in the nucleus [97]. The HBZ-S1 and UBE2D2 PCR primers have been described [46]. *hbz* and *tax* mRNA copy numbers were determined as described [46]. For *hbz* mRNA copies per *UBE2D2* mRNA copy, gene-specific primers were used for cDNA synthesis (HBZ, GCAACCACATCGCCTCCA; UBE2D2, CGTGGGCTCAT AGAAAGCAGTCAA), and a pan-HBZ primer set was used for real-time PCR [CATCGCCT CCAGCCTCCCCT (forward), GAGCAGGAGCGCCGTGAGCGCAAG (reverse)]. This approach was required, as copy number was determined using pSG-THU as a standard [46], which contains the amplicon sequences for the Tax, UBE2D2 and pan-HBZ primer sets, but not for the HBZ-S1 primer set. For *tax* mRNA copies per *UBE2D2* mRNA copy, oligo(dT) primers were used for *tax* and *UBE2D2* cDNA synthesis. Primers for *MYOF* were CAAGCT GATCTCCCTGCTAAA (forward) and ACCTGTCTTCATCACCTTCATC (reverse). Real-time PCR was performed using iTaq Universal Supermix (Bio-Rad) and a CFX Connect Real-Time PCR Detection System (Bio-Rad), and relative mRNA levels were determined as described [46]. Serial dilutions of an appropriate experimental sample were used to generate standard curves for all primer sets included on a PCR plate. From the compilation of all the standard curves for all primers and all PCR plates (analyses), including ChIP PCR plates, the amplification efficiencies ranged from 89–126% with correlation coefficients ranging from 0.942–0.999.

### Cell treatment, transfection and western blotting

WJ460 (DMSO) was purchased from Glixx Laboratories Inc. or MedChem Express; A-485 (DMSO), vacuolin-1 (DMSO) and apilimod (DMSO) were purchased from MedChem Express; EGA (DMSO), leupeptin (H2O) and pepstatin A (DMSO) were purchased from MilliporeSigma; E64D (DMSO) was purchased from Cayman Chemical. Compounds were purchased in solution or reconstituted and were aliquoted and stored at -20˚C or -80˚C as recommended by the manufacturers. Viability following treatment with WJ460 was assessed using alamarBlue (Bio-Rad) according to the manufacturer's instructions. Cells were transfected using Turbofect (Thermo Fisher Scientific) according to the manufacturer's instructions and harvested 24 h later. Whole cell extracts were prepared, and western blotting was done as described [98]. Affinity enrichment of HBZ using GST-KIX was done as described [34]. Antibodies used were as follows: anti-HTLV-1 Env (ARP-1578) and anti-Tax (hybridoma 168B17-

46-92) were obtained from NIH AIDS Research and Reagent Program; anti-HBZ has been described [99]; anti-6x His (ab9108) was purchased from Abcam; anti-MyoF (HPA014245) and anti-Myc (06–340) were purchased from MilliporeSigma; anti-HTLV-1 Env gp46 (1C11, sc-53890; 65/6C2.2.34, sc-57865), anti-Gag p19 (TP7, sc-57870) and anti-β-actin (C4, sc-47778) were purchased from Santa-Cruz; anti-clathrin (D3C6, #4796), anti-caveolin-1 (D46G3, #3267), anti-EEA1 (C45B10, #3288), anti-Rab5a (E6N8S, #46449), anti-Rab7 (D95F2, #9367), anti-Rab11 (D4F5, #5589), anti-LAMP-1 (D2D11, #9091) were purchased from Cell Signaling; anti-EHD2 (PA5-80576) was purchased from ThermoFisher Scientific. All blots were developed using Pierce ECL 2 (Thermo Fisher Scientific) and scanned with a Typhoon RGB imager (Cytiva). Immunoblot quantification was performed using ImageQuant TL v8.1 (Cytiva). To enrich Env SU/gp46 when transiently expressed in HEK293T cells, whole cell extracts were immunoprecipitated with Lens Culinary Agglutinin lectin bound to agarose beads (Vector Laboratories) [26] and beads resuspended in SDS loading dye.

## Adhesion assays

CHO and CHO-LFA1 cells were seeded in 24-well plates at $3\times10^5$ cells/well and cultured overnight. SLB-1 cells and SLB-1 shMYOF and shGFP cells were normalized to $1\times10^6$ cells/mL; SLB-1 cells were treated with WJ460 or DMSO. The infected cells were cultured overnight and then centrifuged and resuspended in PBS to $1 \times 10^6$ cells/mL. Calcein-AM (Thermo Fisher Scientific) was added to the infected cells to a final concentration of 2.5 μM, and the cells were incubated at 37˚C for 20m. During and after Calcein-AM staining, manipulations were done in the dark. Cells were centrifuged and resuspend in fresh, pre-warmed, supplemented IMDM to $1 \times 10^6$ cells/mL and incubated at 37˚C for 10m. The stained cells were then added to the CHO and CHO-LFA-1 cells at $6.4\times10^5$ cells/well, and co-cultures were incubated at 37˚C for 1h. Unattached cells were removed, and wells wash three times with PBS at 37˚C with rocking for 5m. Following aspiration of the final wash, 150 μL of Passive Lysis Buffer (Promega) was added to each well. Cells were lysed on a mechanical shaker for 10m. Lysates were then transferred to 96-well plates and scanned at 488/520 nm using a Varioskan LUX and its associated SkanIt 6.1 software (ThermoFisher Scientific).

## Small RNA interference

The siGENOME SMART pool M-003486-04-0005 and M-003477-02-0005 were used to knock-down p300 and CBP respectively, while the siGENOME Non-Targeting siRNA pool#1 D-001206-13-05 was used as a control (Dharmacon). Cells were seeded to reach ~50% confluence on the day of transfection. Cells were transfected with 25 nM of siRNA using DharmaFECT 1 siRNA transfection reagent (Dharmacon) according to the manufacturer's instructions. The medium was changed 24 h after transfection, and cells were cultured for an additional 48 h prior to collect the total RNA and the cells.

## Chromatin immunoprecipitation (ChIP) assays

ChIP assays were performed using the Zymo-Spin ChIP Kit (Zymo Research) according to the manufacturer's instructions but with certain modifications. For p300 and CBP immunoprecipitations, chromatin was crosslinked using 10 mM disuccinimidyl glutarate (Thermo Scientific) for 45m and then crosslinked with formaldehyde; for all other immunoprecipitations, only formaldehyde was used. Crosslinked chromatin was sonicated using a Misonix Sonicator 4000 (20s pulse on, 30s pulse off, amplitude 40, 5m processing time). Each immunoprecipitation reaction contained 5 μg of antibody and 200 μg of crosslinked, sonicated chromatin. Antibodies used were as follows: anti-p300 (C-20, sc-585) from Santa Cruz Biotechnology; anti-CBP

(D6C5, #7389), anti-JunB (C37F9, #3753) and anti-c-Jun (60A8, #9165) from Cell Signaling Technology; anti-JunD (PA1-834) from ThermoFisher Scientific; anti-MafG (ab154318) from Abcam. HBZ was immunoprecipitated through its C-terminal 6x His tag using an anti-6x His antibody (Abcam, ab9108). Purified ChIP DNA was amplified in iTaq Universal Supermix (Bio-Rad) using a CFX Connect Real-Time PCR Detection System (Bio-Rad). Primer sequences are as follows: MYOF Distal-F, 5'-GAGAAACTTACCAGCCGTTCT; MYOF Distal-R, 5'- AACTACTATTATTACTTGCCTTGGG; MYOF Proximal-F, 5'- CTGGCTCCTGC GTCTAATTT; MYOF Proximal-R, 5'- AATGTCCTGAAGAACGACTTGA; MYOF off-target-F, 5'- GAGCAGGACATGAAGGGAATTA; MYOF off-target-R, 5'-GCTCTATTCAA ACGGCAACAC; WEE1-AP1-F, 5'- CCAATCGGCTTATCGGCTTAT; WEE1-AP1-R, 5'- ACAGGAGCGTGTTTAGGTATTG. Standard curves were generated for primer sets using 5-fold serial dilutions of each input DNA from the ChIP procedure and were included on each experimental plate. Enrichment values were quantified relative to the input as described [100,101]. For 6x His ChIPs, values for the MYOF proximal and distal amplicons were normalized to the value for the off-target amplicon, which was set to 1.

## Infection assays

Infected cells ($1 \times 10^5$) were co-cultured with Jurkat pminLUC-viral CRE cells ($2 \times 10^5$) for 24h. In experiments using WJ460, infected cells were pretreated with WJ460 or DMSO for less than 24h, then washed twice with medium prior to co-culture. In experiments using CHOK1-Luc cells as target cells, $1 \times 10^5$ cells were plated per well in 24-well plates, cultured for 24h, and then exposed to $5 \times 10^5$ infected cells/well for 1.5h. Wells were then washed four times with PBS to remove effector cells, and the remaining CHOK1-Luc cells were cultured in supplemented DMEM for an additional 24h. For some experiments, infected cells were irradiated (77 Gy) using a MultiRad 350 X-Ray Irradiator before incubation with the target cells. Cells were lysed with 100 μL of Cell Culture Lysis Reagent or Passive Lysis Buffer (Promega), samples were normalized according to total protein, and luciferase activity was measured using the Luciferase Assay System (Promega) and a GloMax 20/20 Luminometer (Promega).

## Immunofluorescence microscopy

Cells were treated with pepstatin A (15 μg/mL) and leupeptin (25 μg/mL) for 4h and then washed and suspended in unsupplemented RPMI and seeded at $7.2 \times 10^5$ cells/chamber in 8 well μ-Slides (ibidi) coated with poly-L-lysine. Protease inhibitors were used to limit Env degradation in lysosomes, thereby enhancing Env detection in these organelles. Cells were incubated at 37°C for 15m and then fixed with cold PBS (137 mM NaCl, 2.7 mM KCl, 3 mM Na$_2$HPO$_4$, and 1.5 mM KH$_2$PO$_4$)/4% paraformaldehyde on ice for 30m. Cells were washed with PBS and permeabilized and blocked with PBS/0.1% saponin/5% goat serum for 1h at room temperature. Cells were then probed with 1:50 antibody dilutions of Alexa Fluor 594-conjugated anti-LAMP-2 (H4B4, Santa Cruz Biotechnology, sc-18822 AF594) and Alexa Fluor 488-conjugated HTLV-1 gp46 (1C1, Santa Cruz Biotechnology, sc-53890 AF488) in PBS/0.1% saponin/1% BSA overnight at 4°C. Cells were then washed with PBS and overlayed with ibidi Mounting Medium. Fluorescence images were acquired by confocal microscopy using an LSM 700 microscope (Zeiss), and images were analyzed using Fiji (ImageJ, version 1.53t). Specifically, masks were used to define cells as regions of interest (ROIs) and to show cell boarders. Masks were established from threshold-adjusted Z-projections of max intensities summed from both channels. Colocalization was analyzed with the JaCoP plugin [102]. Prior to colocalization analysis, ROIs were subjected to background subtraction and deconvolution.

Colocalization was performed on a single slice from each ROI that contained a high proportion of pixels displaying above-background fluorescence for each channel with priority placed on the gp46 channel.

## Flow cytometry

Cells were equalized to 5 x $10^5$ cells/mL and cultured for 24h prior to analysis. A total of 5 x $10^5$ cells/labeling reaction were collected by centrifugation at 800 x g for 3 min at 4°C, washed once in 2 mL of cold PBS/0.2% BSA (FACS buffer), then suspended in 50 μL of cold FACS buffer to which 2 μg of primary antibody was added. Antibodies used were as follows: anti-ICAM-1 (P2A4, Millipore Corporation) and anti-HTLV-1 gp46 (1C11, sc-53890, Santa Cruz Biotechnology). Cells were labeled on ice for 1 h and then washed with 2 mL of FACS buffer. For ICAM-1 labelling, cell pellets were suspended in 50 μL of FACS buffer to which 0.25 μg of fluorescein isothiocyanate (FITC) goat anti-mouse Ig (Southern Biotech) was added. For gp46 labelling, 0.25 μg of Allophycocyanin (APC) goat anti-mouse Ig (Southern Biotech) was used. Cells were labeled on ice for 30 m, washed with 2 mL of FACS buffer, and fixed with 2% para-formaldehyde (PFA) at 4°C for at least 30 m. Cells were suspended in 500 μL FACS buffer, and analyzed using Cytek Aurora flow cytometer (Cytek Biosciences). Data were analyzed using FlowLogic Software.

## Detection of Env SU/gp46, virion and Gag p19 ELISA in the culture medium

To precipitate proteins from culture media, cells were washed in serum-free IMDM, seeded at $2.5x10^6$ cells/mL in serum-free IMDM and cultured for 24h. Cells were pelleted by centrifuged at 1000 x g for 3m, and supernatants were passed through 0.45 μm PES filters. Ice-cold 100% trichloroacetic acid (TCA) was added to a final concentration of 20%, samples were vortexed, and chilled on ice overnight. Precipitated proteins were pelleted by centrifugation at 10,000 x g/4°C for 5m. Protein pellets were washed three times with ice-cold 10 mM HCl/90% acetone and then resuspended in SDS loading dye. To concentrate cell-free virus particles, cells were seeded at $1.5x10^6$ cells/mL in supplemented IMDM and cultured for 24h. Culture media were then clarified as described above and centrifuged in a SW 40 Ti rotor (Beckman Coulter) at 20,000 rpm/4°C for 2h. Supernatants were removed, and virus was suspended in 250 uL of STE (0.1 M NaCl, 10 mM Tris, 1 mM EDTA, pH 7.5). Virus was heat-inactivated and stored at -80°C until western blot analysis. For ELISA detection, HTLV-1 infected cells were washed twice, in some experiments, treated with WJ460, and incubated for 24 h in serum-free IMDM (pilot experiments verified that WJ460 produced the same effects under this condition). Supernatants were collected, passed through 0.45 μm PES filters, and Gag p19 levels were measured by ELISA according to the manufacturer's instructions (ZeptoMetrix Corporation). Absorbances were detected with an accuSkan FC (Fisher Scientific).

## *In silico* analysis and statistical analysis

Microarray data sets used in this study are available at NCBI Gene Expression Omnibus (GEO): accession numbers GSE29312 [103]; GSE14317 [104] and GSE94409 [56]. For each sample, probes corresponding to the MYOF transcript were identified and GEO2R was used to obtain expression values. ChIP-Seq data sets from GEO accession number GSE94732 [56] were analyzed using the Human Mar. 2006 (NCBI36/hg18) assembly with the IGV Browser [105]. Two-tailed Student's t-tests were used for two-group comparisons and significance was established at $p < 0.05$.

## Supporting information

**S1 Fig. Tax does not affect the level of MyoF expression in HeLa cells. (A)** MyoF expression in HeLa cells ($2.5 \times 10^5$ cells) transiently transfected with 4 μg pSG5 or pSG-Tax-His using TurboFect. Whole cell extracts (50 μg) were analyzed 48h later by Western blot using antibodies against MyoF, Tax (6xHis) and β-actin. **(B)** The graph shows quantification of band intensities of MyoF normalized to band intensities of β-actin averaged from four independent experiments. Error bars show standard deviations.
(TIF)

**S2 Fig. *hbz* and *tax* mRNA copy numbers compared to relative *MYOF* mRNA levels in HTLV-1-infected T-cell lines. (A)** and **(C)** *hbz* mRNA and *tax* mRNA copies per *UBE2D2* mRNA copy (housekeeping gene), respectively, in HTLV-1-infected T-cell lines. **(B)** and **(D)** Linear regression analysis of relative *MYOF* mRNA to *hbz* mRNA copies and *tax* mRNA copies, respectively. *MYOF* mRNA values were normalized to that for SLB-1 cells (set to 1). A Spearman correlation test was used based on the none-normal distribution of the data.
(TIF)

**S3 Fig. Relative *hbz* and *tax* mRNA levels compared to relative *MYOF* mRNA levels in recently established HTLV-1-immortalized clones.** *MYOF*, *hbz* and *tax* mRNA levels were normalized to that for clone C15 (set to 1). A Spearman correlation test was used based on the none-normal distribution of the data.
(TIF)

**S4 Fig. DNA sequences corresponding to the distal and proximal peaks of HBZ-enrichment within the *MYOF* gene.** Peak sequences were determined using the UCSC Genome Browser (http://genome.ucsc.edu; [79]) with peak coordinates subsequently extrapolated to the current genome assembly. Bolded sequences denote full or partial AP-1 binding sites.
(TIF)

**S5 Fig. Cell viability following treatment with WJ460. (A)** ATL-2 cells were treated with DMSO or WJ460 (200 nM and 1μM) for 24 h. **(B)** ATL-2 and SLB-1 cells were treated with DMSO or 1μM WJ460 for 24 h. Graphs show average cell viabilities for eight (A) and six (B) replicates. Values are normalized to a single DMSO treatment replicate for each experiment/cell line.
(TIF)

**S6 Fig. Targeting MyoF reduces HTLV-1 infection. (A)** SLB-1 cells were co-cultured with Jurkat-pminLUC-vCRE cells, harvested and analyzed for luciferase activity at the times indicated. The graph shows luciferase assay results average from three replicates for each time point and is representative of two independent experiments. **(B)** SLB-1 cells were treated with DMSO or 1 μM WJ460 prior to co-culture with CHOK1-Luc cells. **(C)** ATL-2 were treated with DMSO or 1 μM WJ460 prior to co-culture with Jurkat-pminLUC-vCRE. **(D)** The HTLV-1-immortalized primary human T-cell clone, CJ4, was treated with DMSO or 1 μM WJ460 prior to co-culture with Jurkat-pminLUC-vCRE cells. **(E)** ATL-2 cells stably expressing an shRNA targeting GFP or *MYOF* mRNA were co-cultured with CHOK1-Luc cells. **(F)** SLB-1 cells were treated with DMSO or 1 μM WJ460 and then irradiated (77 Gy) prior to co-culture with Jurkat-pminLUC-vCRE cells. **(G)** ATL-2 cells stably expressing an shRNA targeting GFP or *MYOF* mRNA were irradiated (77 Gy) prior to co-cultured with Jurkat-pminLUC-vCRE cells. Graphs show luciferase assay results averaged from three replicates for each condition of a single experiment; $^{**}p<0.01$, $^{***}p<0.001$. Graphs (B) and (C) are representative of at least three independent experiments. Graphs (A), (D), (E), (F) and (G) are representative of two

independent experiments.
(TIF)

**S7 Fig. Inhibition of lysosomal proteases partially restores intracellular levels of SU (gp46) in HTLV-1 infected cells treated with WJ460.** ATL-2 cells were treated with DMSO or 1 μM WJ460 with or without 5 mg/mL leupeptin (L) and 5mg/ml pepstatin A (P) or 25 μM E64D (E) for 24 h. Whole cell extracts (50 μg) were analyzed by Western blot using antibodies against β-actin and SU/Pr.
(TIF)

## Acknowledgments

We would like to thank Rikinari Hanayama for providing the myoferlin plasmid.

## Author Contributions

**Conceptualization:** Nicholas Polakowski, Isabelle Lemasson.

**Data curation:** Isabelle Lemasson.

**Formal analysis:** Nicholas Polakowski, Md Abu Kawsar Sarker, Kimson Hoang, Isabelle Lemasson.

**Funding acquisition:** Patrick L. Green, Isabelle Lemasson.

**Investigation:** Nicholas Polakowski, Md Abu Kawsar Sarker, Kimson Hoang, Georgina Boateng, Amanda W. Rushing, Wesley Kendle, Amanda R. Panfil, Isabelle Lemasson.

**Methodology:** Nicholas Polakowski, Amanda R. Panfil, Isabelle Lemasson.

**Project administration:** Isabelle Lemasson.

**Resources:** Claudine Pique, Patrick L. Green, Amanda R. Panfil, Isabelle Lemasson.

**Supervision:** Nicholas Polakowski, Isabelle Lemasson.

**Validation:** Nicholas Polakowski, Md Abu Kawsar Sarker, Kimson Hoang, Isabelle Lemasson.

**Visualization:** Nicholas Polakowski, Isabelle Lemasson.

**Writing – original draft:** Nicholas Polakowski, Isabelle Lemasson.

**Writing – review & editing:** Nicholas Polakowski, Claudine Pique, Patrick L. Green, Amanda R. Panfil, Isabelle Lemasson.

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
