## [Decision Letter · Decision Letter 0]

5 Jan 2023

Dear Dr. Lemasson,

Thank you very much for submitting your manuscript "HBZ upregulates myoferlin expression to facilitate HTLV-1 infection." for consideration at PLOS Pathogens. As with all papers reviewed by the journal, your manuscript was reviewed by members of the editorial board and by several independent reviewers. The reviewers appreciated the attention to an important topic. Based on the reviews, we are likely to accept this manuscript for publication, providing that you modify the manuscript according to the review recommendations.

The reviewers agreed that your findings are of interest and importance in understanding HTLV-1 cell-to-cell transmission. However, both reviewers have made several suggestions for clarification or minor modification of the manuscript. In your revised manuscript, please indicate clearly how you respond to each of the points raised.

Sincerely,

Charles R. M. Bangham

Academic Editor

PLOS Pathogens

Richard Koup

Section Editor

PLOS Pathogens

Kasturi Haldar

Editor-in-Chief

PLOS Pathogens

orcid.org/0000-0001-5065-158X

Michael Malim

Editor-in-Chief

PLOS Pathogens

orcid.org/0000-0002-7699-2064

The reviewers agreed that your findings are of interest and importance in understanding HTLV-1 cell-to-cell transmission. However, both reviewers have made several suggestions for clarification or minor modification of the manuscript. In your revised manuscript, please indicate clearly how you respond to each of the points raised.

Reviewer Comments (if any, and for reference):

Reviewer's Responses to Questions

**Part I - Summary**

Reviewer #1: In this manuscript, Polakowski et al. identified Myoferlin (MyoF) as a novel host factor with importance for HTLV-1 transmission. Specifically, the authors found expression of MyoF in a panel of HTLV-1 infected cell lines and in published microarray data obtained from HTLV-1 infected patients. They confimed their findings also in experimentally-infected cell clones. Using HBZ-expressing HeLa clones, the authors found that HBZ is able to induce MyoF gene expression, which was further corroborated by data from publically available arrays in ATL-cell lines upon knockout of HBZ. Using ChIP assays, the authors found binding of HBZ either to two or one site far downstream of the MyoF transcriptional start site, which was accompanied by binding of Jun B and c-Jun, but not JunD. Further ChIP assays and use of specific siRNAS or inhibitors revealed that HBZ activates MyoF transcription depending on p300/CBB activity. Co-culture experiments showed that inhibition or shRNA-mediated repression of MyoF in HTLV-1 infected donor cells reduces transactivation of co-cultured reporter CHOK1-Luc or Jurkat-pmini-LUC-vCRE cells, suggesting that MyoF contributes to HTLV-1 cell-to-cell transmission. These findings were strengthened by data showing that MyoF inhibition reduces the abundance of intracellular Env gp46, while expression of the Env precursor and surface expression of Env were unaffected. However, authors showed that less Env is incorporated into virions upon repression of MyoF. Mechanistically, authors provide evidence that MyoF restricts lysosomal degradation of Env, leading to endosomal recycling of Env and Env incorporation into viral particles.

This is a very interesting study which provides not only detailed insights into transcriptional regulation of MyoF by HBZ, but which also uncovers how MyoF impacts HTLV-1 transmission. The manuscript is well-written and the experiments are well-designed. Authors embed publically available data sets in their study, which is an elegant approach to make large public datasets usable in a resource-saving way.

I have some comments to improve the quality of this manuscript.

Reviewer #2: Polakowski et al. describe a study demonstrating that HTLV-1 HBZ upregulates expression of the gene coding for myoferlin (MYOF), a protein involved in membrane dynamics and vesicle transport. They show that MYOF induction is driven by formation of complexes between HBZ and Jun family members and recruitment of p300/CBP to enhancer elements. Experiments with a drug that inhibits MyoF and shRNA targeting MyoF showed that the protein regulates cell adhesion properties, HTLV-1 infection efficiency and intracellular trafficking of the viral envelope protein.

The identification of the role of HBZ in enhancing infection through regulation of MYOF is an important finding and adds to the understanding of factors controlling virus transmission. The study is sound and the descriptions are clear.

With minor modifications, the manuscript would be suitable for publications.

**Part II – Major Issues: Key Experiments Required for Acceptance**

Reviewer #1: Figure 1C: it would be benefitial to present Tax and HBZ copy numbers of the HTLV-1-infected cell lines and to present correlations with MyoF transcripts.

Figure 1D: Can HBZ protein be detected in these cell lines?

Figure 2B, Lines 187-188: please show correlations between HBZ expression and Tax expression, respectively, with MyoF (e.g. in supplement).

Figure 5: The most important control of the experiments shown in 5 are co-cultures between non-producing C8166 cells and the reporter cells. Please include this control in the main manuscript and not only on the supplement.

Reviewer #2: On lines 187-188, the Authors state that there was no correlation between Tax and MOYF expression levels in their infected clones, and that this confirms that Tax is not involved in MYOF induction. This assertion should be supported by experiments in which MYOF is quantified by immunoblot in Tax-transfected cells (HeLa cells or a Tax-negative T-cell line).

**Part III – Minor Issues: Editorial and Data Presentation Modifications**

Reviewer #1: Figure 1B: Please show HBZ protein expression.

Figures 1 E, F, 2A, 3A: These data were obtained from published data sets. Please provide reference of data set also in the respective figure legends.

Figure 5D: This experiments was just performed twice, therefore, I would remove statistics from this experiments.

Earlier work has shown that transactivation of co-cultured reporter cells peaks at 2-4 d post co-culture (PMID: 26269171). Is this also true for the reporter cell lines used by the authors?

Line 198: TSS should be introduced

Line 556: LRS should be introduced

Line 581: is there a reason why Olido (dT) primers instead of random hexamer primers were used for cDNA synthesis prior to HBZ qPCR? Please explain.

Line 671: please correct: „were co-cultured“

Figure 7E/ Line 684: Please explain: why are cells treated with protease inhibitors prior to imaging? To prevent Env degradation?

Line 713: typo: precipitate instead of precipitated

Line 735: reference is missing

Reviewer #2: The immunoblots shown in Fig. 6A-C should be accompanied by bar graphs showing differences in levels of proteins of interest (normalized against beta-actin) calculated as means from at least 3 experimental replicates. From the immunoblots shown in Fig. 6D-F, it appears that MYOF induces expression of PR/gp62; the increase in SU/gp46 is much less evident.

Lines 695-696 and legend to Fig. 7E: more detail on confocal microscopy should be provided. Was colocalization calculated layer-by-layer? What was the thickness of the optical slice? How were the borders of the cells visualized?

Was the antibody used to detect Su/gp46 in immunofluorescence and flow cytometry specific for this protein, or does it also detect Pr/gp62?

Abstract, lines 27-31. The two phrases describing HBZ’s enhancement of infection through upregulation of ICAM-1 could be replaced by a single phrase similar to the one on lines 100-101; this would provide a smoother transition into the topic of the study.

Abstract, line 37: the text should indicate that WJ460 is a drug.

In Figure 1C, the label ‘ATL patients’ should be replaced by ‘ATL-derived’ to make it clear that these were not patient samples. The figure legend (line 1110) should read ‘ATL-derived cell lines’, and provide the references for the published datasets (lines 1115-1116).

On line 170, replace ‘ATL patient cells’ with ‘ATL cell lines’. The legend to Figure 2A should indicate the control used to calculate the percent reduction in MYOF transcript levels.

Line 247: delete ‘Fig.2.’

In Fig. 6a, there might have been an error in image assembly - the immunoblot strip for Env proteins appears to show several images superimposed over each other.

The colors in the graphs in Figs. 7C and 8A do not correspond to the legends.

On line 735, the reference for GSE94409 is missing.

Line 747: References

Lines 1191-1193, 1201-1204, 1281-1282: do the graphs show the means from 3 experiments? Or is this a single experiment that is representative of a total of 3 experiments? (Also see line 1335, ‘presentative’). In general, the exact number of experiments analyzed should be indicated, rather than saying ‘at least 3’.

Antibody C4, sc-47778 recognizes beta-actin. This should be indicated in the text (line 607) and in the immunoblot figures.

PLOS authors have the option to publish the peer review history of their article (what does this mean?). If published, this will include your full peer review and any attached files.

Reviewer #1: No

Reviewer #2: No

Figure Files:

Data Requirements:

Reproducibility:

References:

---

## [Editor Report · Decision Letter 1]

10 Feb 2023

Dear Isabelle,

We are pleased to inform you that your manuscript 'HBZ upregulates myoferlin expression to facilitate HTLV-1 infection.' has been provisionally accepted for publication in PLOS Pathogens.

Best regards,

Charles

Charles R. M. Bangham

Academic Editor

PLOS Pathogens

Richard Koup

Section Editor

PLOS Pathogens

Kasturi Haldar

Editor-in-Chief

PLOS Pathogens

orcid.org/0000-0001-5065-158X

Michael Malim

Editor-in-Chief

PLOS Pathogens

orcid.org/0000-0002-7699-2064
---

## [Editor Report · Acceptance letter]

20 Feb 2023

Dear Dr. Lemasson,

We are delighted to inform you that your manuscript, "HBZ upregulates myoferlin expression to facilitate HTLV-1 infection.," has been formally accepted for publication in PLOS Pathogens.

Best regards,

Kasturi Haldar

Editor-in-Chief

PLOS Pathogens

orcid.org/0000-0001-5065-158X

Michael Malim

Editor-in-Chief

PLOS Pathogens

orcid.org/0000-0002-7699-2064